

# Satellite-observed sea ice area flux through Baffin Bay:1988-2015

Haibo Bi[1,2,3], Yunhe Wang[1,2], Xiuli Xu[1,2,3], Yu Liang[1,2], Jue Huang[5], Yilin Liu[5], Min Fu[6], Haijun Huang[1,2,3,4]

[1]Key laboratory of Marine Geology and Environment, Institute of Oceanology, Chinese Academy of Sciences, Qingdao, China

[2]Laboratory for Marine Geology, Qingdao National Laboratory for Marine Science and Technology, Qingdao, China

[3]Center for Ocean Mega-Science, Chinese Academy of Sciences, Qingdao, China

[4]University of Chinese Academy of Sciences, Beijing, China

[5]Shandong University of Science and Technology, Qingdao, China

[6]Key Laboratory of Research on Marine Hazards Forecasting, National Marine Environmental Forecasting Center, Beijing,
China

*Correspondence to*: Haijun Huang (hjhuang@qdio.ac.cn)

**Abstract.** Sea ice export through Baffin Bay plays a vital role in modulating the meridional overturning process in the downstream Labrador Sea. In this study, satellite-derived sea ice products are explored to obtain the sea ice flux (SIF) through three passages (referred to as A, B, and C for the north, middle, and south passages, respectively) of Baffin Bay.

Over the period 1988-2015, the average annual (October-September) sea ice area export is $555 \times 10^3 \, km^2$, $642 \times 10^3 \, km^2$, and $551 \times 10^3 \, km^2$ through passages A, B, and C, respectively. These amounts are less than that observed through the Fram Strait (FS, $707 \times 10^3 \, km^2$). Clear increasing trends in annual sea ice export on the order of $53.1 \times 10^3 \, km^2/de$ and $43.2 \times 10^3 \, km^2/de$ are identified at passages A and B, respectively. The trend at the south passage (C), however, is slightly negative $(-13.3 \times 10^3 \, km^2/de)$. The positive trends in annual SIF at A and B are primarily attributable to the increase during winter

months, which is triggered by the accelerated sea ice motion (SIM) and partly compensated by the reduced sea ice concentration (SIC). During the summer months, the sea ice export through each Baffin Bay passage usually presents a negative trend, primarily because of the decline in SIM and it is further enhanced by a dramatic decrease in SIC. A significant positive trend in the net SIF (i.e. net ice inflow) is found for between the passages A (or B) and C at 54.5 (or 64.2) $\times 10^3 \, km^2/de$. Therefore, Baffin Bay may have presented a greater convergence of ice. Overall, the connection between

Baffin Bay sea ice export and the North Atlantic Oscillation (NAO) is tenuous, although the correlation is sensitive to variations in the selected time period. In contrast, the association with the cross-gate sea level pressure difference (SLPD) is robust in Baffin Bay (R = 0.69 - 0.71 depending on the passages), but relatively weaker compared with that in the FS (R = 0.74). Baffin Bay is bounded by Baffin Island to the west and Greenland to the east, thus, sea ice drift is not converted to the free state observed in the FS.



## 1. Introduction

Baffin Bay is a semi-enclosed ocean basin between Baffin Island and Greenland that connects the Arctic Ocean and the Northwest Atlantic. Sea ice outflow through Baffin Bay is a key component of the freshwater balance of the Labrador Sea (the northwest arm of the North Atlantic) further downstream, and the anomalous outflows of sea ice contribute to the freshening of surface waters of the Labrador Current (Goosse et al., 1997; Curry et al., 2014; Qian et al., 2016). When higher outflows of lighter, fresher melt water enter the Labrador Sea, they can strengthen ocean stratification and thus have a potential impact on the convective overturning process of water masses (Aagaard and Carmack, 1989; Hawkins et al., 2011; Cimatoribus et al., 2012). As a result, the thermohaline circulation process in the North Atlantic Ocean could be suppressed (Holland et al., 2001; Jahn et al., 2010). Thus, variations of surface fresh water content related to the sea ice outflow from Baffin Bay presents one of the driving forces for the "global ocean conveyor belt" in the Labrador Sea (Rudels, 2010; Qian et al., 2016). This process is a vital for the transport of heat and salt into northern latitudes. Moreover, the North Atlantic ecosystem and fisheries may be impacted by changes in the sea ice flux (SIF) through Baffin Bay, which may alter the physical properties of productive coast banks and slope areas (Hansen et al., 2003; Drinkwater, 2009).

A strong atmospheric warming trend has been observed in the Arctic (Serreze et al., 2009; Stroeve et al., 2014; Graham et al., 2017; Stroeve et al., 2018), involving Baffin Bay (2 to 3 ℃/de) since the late 1990s (Peterson and Pettipas, 2013). Prolonged days of sea ice melting (i.e. earlier melting onset and delayed ice-freezing startup) have been observed around Baffin Bay (Stroeve et al., 2014). The coincident rapid decline of sea ice cover in all seasons is also observed in the bay (Comiso et al., 2017b; Parkinson and Cavalieri, 2017). Within the context of such a radical climate change, examining the variability and trends of sea ice flux (SIF) through Baffin Bay is of particular interest.

Compared with the sparsely available in-situ measurements, satellite observations are unique in their ability to continually capture the full spatiotemporal information of the polar sea ice cover over past decades (Comiso and Hall, 2014; Parkinson, 2014; Tilling et al., 2016; Parkinson and Cavalieri, 2017). Sea ice export has been investigated in several key water fluxgates around the periphery of the Arctic Basin (such as the Fram Strait (FS), Laptev Sea, Nares Strait, etc.) (Kwok et al., 2005; Spreen et al., 2006; Kwok, 2007; Kwok, 2009; Kwok et al., 2010; Smedsrud et al., 2011; Krumpen et al., 2013; Kwok et al., 2013; Bi et al., 2016a; Bi et al., 2016b; Krumpen et al., 2016; Smedsrud et al., 2017; Zhang et al., 2017), while the FS has been a primary focus owing to its linkage to Arctic sea ice extent (Smedsrud et al., 2011; Smedsrud et al., 2017). Sea ice transport via the Davis Strait (southern fluxgate of the Baffin Bay) was also previously examined, but only for several short episodes (see Figure 2 for details) owing to the limited available data (Cuny et al., 2005; Kwok, 2007; Curry et al., 2014), and the trends and variability of sea ice exported out of Baffin Bay over the past several decades are poorly understood. Here, satellite-derived sea ice products are used to establish a 27-year record of SIF estimates (1988-2015) at

three passages in Baffin Bay (Figure 1). The SIF variability and trends are examined in detail, and their relationships with typical climate factors as well as atmospheric forcing are further assessed.

This study is organized as follows. Section 2 describes the data and methodology used to calculate the SIF and provides its uncertainty estimate. Section 3 presents the variability and trends in the SIF fields at the three passages as well as a

comparison with the SIF of the FS is depicted in this section. Section 4 outlines the connections of the SIF through Baffin Bay with climate factors (surface winds (SW), sea level pressure (SLP), surface air temperature (SAT), and sea surface temperature (SST)), and discusses the linkage with preferred atmospheric forcings, including the North Atlantic Oscillation (NAO) and cross-gate sea level pressure difference (SLPD), is discussed. Section 5 concludes the study.

## 2. Data and methodology

### 2.1 Data

### 2.1.1 Sea ice motion

One of the most comprehensive sea ice motion (SIM) datasets is provided by the National Snow and Ice Data Center (NSIDC) (Tschudi et al., 2016). This product has contributed to a number of studies associated with polar sea ice and has been widely employed among the modeling and data assimilation communities (http://nsidc.org/data/NSIDC-0116). This

dataset is derived from a variety of satellite-based sensors, including the Advanced Very High Resolution Radiometer (AVHRR), the Scanning Multichannel Microwave Radiometer (SMMR), Special Sensor Microwave Imager (SSM/I), the Special Sensor Microwave Imager Sounder (SSMIS), and the Advanced Microwave Scanning Radiometer-Earth Observing System (AMSR-E), merged with buoy measurements from the International Arctic Buoy Program (IABP) and estimates determined from the reanalyzed wind data.

The Polar Pathfinder Daily 25 km EASE-Grid Sea Ice Motion Vectors is obtained for the period from 1978 to 2015 (Tschudi et al., 2016). However, the satellite scanner on the satellite SMMR was operated only on alternate days due to spacecraft power limitations, and earlier observational data were only collected every other day for the period October from 1978 to August 1987. In this case, half of the daily satellite-derived SIM as well as sea ice concentration (SIC) records are dismissed, which hampers the ability to obtain an accurate estimate for the monthly SIF before 1987. After the launch and

operation of the SSM/I since July 1987, daily satellite observations became available. In addition, previous studies reported that the lower SIF estimate in FS area based on the NSDIC SIM before 1987 seems to be unrealistic (Bi et al., 2016b; Smedsrud et al., 2017), which suggests a noticeable discontinuity between the NSDIC sea ice motion record based on observations from the SMMR and subsequent SSM/I series. To reduce the uncertainty caused by the shifting satellite sensors, we use only the NSIDC daily SIM data for the period from 1988-2015.





### 2.1.2 Sea ice concentration

SIC data (1988-2015) were also obtained from the NSIDC (http://nsidc.org/data/NSIDC-0079). These data are derived from the SMMR onboard the Nimbus-7 satellite, the SSM/I onboard the Defense Meteorological Satellite Program (DMSP) –F8, F11 and F13, and SSMIS aboard DMSP-F17 using the Bootstrap algorithm (Comiso et al., 2017a). This dataset provides an

5 improved consistency between sensors through the use of daily varying tie points generated using the AMSR-E. In addition, the product provides an enhanced removal of weather and land contamination. A climatological SST mask was applied to remove pixels from regions where the ocean surface is above freezing. Additionally, land contamination (false ice along the coast due to pixels containing a mixture of land and ocean) were removed using a filter adapted from Cho et al. (1996). Daily data (1988-2015) used in this study were projected onto the 25-km polar stereographic grids.

**2.1.3 SLPD and NAO**

The SLP is obtained from the National Centers for Environmental Prediction/ National Center for Atmospheric Research (NCEP/NCAR) (https://www.esrl.noaa.gov/psd/cgi-bin/db_search/DBListFiles.pl?did=195&tid=61156&vid=675) reanalysis data (Kalnay et al., 1996). SLPD is calculated as the SLP difference between the western and eastern endpoints of a passage (lines as marked in Figure 1, passage A: (77.1 ˚W, 73.6 ˚N) to (68.0 ˚W, 75.9 ˚N); B: (67.4 ˚W, 70.2 ˚N) to (55.9 ˚W, 71.4 ˚N);

C: (61.1 ˚W, 66.7 ˚N) to (54.1 ˚W, 67.2 ˚N)). The atmospheric structure linked to positive/negative SLPD phases (SLPD+/-) indicates ice export/import through a passage.

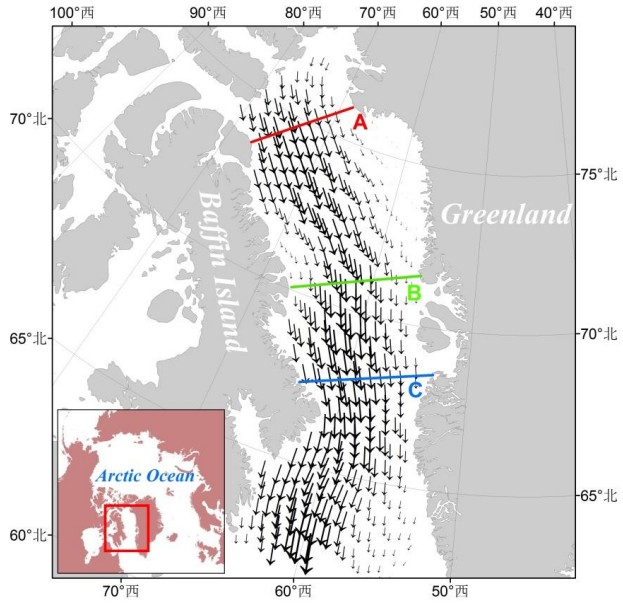





**Figure 1.** Locations of fluxgates used to obtain the SIF over Baffin Bay. Passages A (red line), B (green line), and C (blue line) are positioned in the northern, middle, and southern parts, respectively. The inset map points to the general location of our study area from a broader view. The black arrows denote NSIDC sea ice motion vectors (unit: cm/s).

The NAO index is obtained from NCEP Climate Prediction Center (CPC) (http://www.cpc.ncep.noaa.gov/products/precip/CWlink/pna/nao_index.html). The NAO represents a climatic phenomenon in the North Atlantic Ocean associated with fluctuations between the Icelandic low and the Azores high (Hurrell, 1995). It captures the primary variations (approximately 30%) of the monthly SLP all year round in the north Atlantic regions. Ideally, an intense Icelandic low with a strong Azores ridge to its south leads to a positive NAO mode (i.e., NAO+), which could enhance the sea ice extent in Baffin Bay, whereas a positive SLP anomaly over the Icelandic low during negative phase (NAO-) would reduce the ice extent through the bay (Stern and Heide-Jørgensen, 2003). The connection between the NAO and sea ice transport over the Baffin Bay region remains to be assessed on longer time series observations (Häkkinen and Cavalieri, 2005).

## 2.2 Area flux retrieval, uncertainty estimate, and assessment

The SIF is estimated by taking the integral of the product between the gate-perpendicular component of the SIM and SIC along a gate (Kwok, 2007). To inspect the regional variations of the SIF, three passages are considered (A, B and C as shown in Figure 1), located at the northern, middle, and southern parts of Baffin Bay, respectively. The north gate (A), which is ~370 km wide (red in Figure 1), is positioned at ~73 °N, whereas the south gate (C), which is ~450 km wide (blue in Figure 1), is positioned at ~67 °N, and the middle gate (B), which is ~ 440 km wide (green in Figure 1), is positioned in ~70 °N.

The sea ice area flux across a gate is written as follows:

$$SIF = G \sum_{i=1}^{N-1} u_i c_i (i = 1,2,\dots,N) \qquad (1)$$

where $N$ is the number of grids along the gate and $G$ corresponds to the length (25 km) of a grid, $u_i$ and $c_i$ are SIM and SIC at the $i$-th grid. Passage A spans a length of ~372 km, corresponding to fourteen 25-km grids, while B (~450 km in width) and C (~321 km in width) consist of seventeen and eighteen grids with a 25-km width, respectively. SIM at the endpoints of any passage, about within a narrow zone of 10 km, are constrained to approach zero (Kwok et al., 2004; Kwok, 2009).

**Table 1.** Mean uncertainty estimates in regards to the daily ($\sigma_D$), monthly ($\sigma_m$), and annually SIF ($\sigma_a$) for the period of from 1988-2015.

| Passages | Width (km) | $N_s$ | $\sigma_D$ ($10^3$ km$^2$) | $\sigma_m$ ($10^3$ km$^2$) | $\sigma_a$ ($10^3$ km$^2$) |
|---|---|---|---|---|---|
| A | 372 | 14 | 2.41 | 13.19 | 45.69 |
| B | 442 | 17 | 3.15 | 17.27 | 59.82 |





| C | 453 | 18 | 3.32 | 18.21 | 63.09 |
|---|-----|----|------|-------|-------|

The monthly SIF refers to the cumulative results of the daily flux over a calendar month. Likewise, the annual SIF denotes the cumulative monthly area flux of one year. With the assumption that uncertainty in the SIM samples is additive, unbiased, uncorrelated, and normally distributed (Kwok, 2009), the errors in the daily area flux estimate can be calculated as follows (Kwok, 2009): $\sigma_D = \sigma_u L / \sqrt{N_S}$, where $L$ is the width of a passage, $\sigma_u$ is the uncertainty in daily SIM and $N_s$ is the

number of independent samples across a gate (Table 1). For $\sigma_u$, we use the upper limit of the uncertainty determined through comparisons with buoy drifts, and it corresponds to 1.73 km/day (or 2 cm/s) for the NSIDC SIM data (Sumata et al., 2014). The uncertainty in the monthly area flux estimate is obtained with $\sigma_m = \sigma_D \sqrt{N_D}$, where $N_D$ is the number of days over the month of interest. Thereby, the annual flux uncertainty is calculated as $\sigma_a = \sigma_m \sqrt{N_m}$, where $N_m = 12$ is the number of calendar months from October to the following September, thus representing a complete production-and-decay cycle of sea

ice. The annual export uncertainty $\sigma_a$ (Table 1) corresponds to 8.23%, 9.32%, 11.66% of the average amounts of annual SIF at passages A, B, and C, respectively. (see Sect. 3.3.3 for details).

### 3. Results

### 3.1 Comparison to previous results

Using SSM/I observations, Cuny et al. (2005) obtained the winter (December-May) SIF through the Davis Strait over the

period 1991/1992-1999/2000. The results are presented in Figure 2 (green line). For this decade, the mean winter (November-May) SIF was estimated to $496 \times 10^3\,\mathrm{km}^2$. In comparison, our NSIDC-derived SIF results at passage C (close to Davis Strait) for the corresponding winter at the same period is, on average, slightly lower by $-10.4 \pm 54.6\,\mathrm{km}^2$. The number after $\pm$ denotes the standard deviation of the difference. In addition to the small difference in the passage locations, the moderate correlation (R=0.56) between the two records is mainly due to the distinct contrast in the spatial resolution of the

SIM data utilized (~70km for SSM/I 37 GHz vs 25 km for the NSIDC product), since larger uncertainty is expected in the SIF estimates based on spatially coarser SIM data.

Kwok (2007) used AMSR-E 89 GHz data to examine the sea ice drift and export in Baffin Bay over the period from 2002/2003-2006/2007, and these data are extended to 2004/2005-2009/2010 in Curry et al. (2014). Kwok (2007) compared the SIM from AMSR-E imagery (~ 6 km) with those retrieved from high-resolution (several hundred meters) Envisat SAR

observations in the northern Baffin Bay area. The correlation analysis indicate that AMSR-E SIM accounted for approximately 90% of the variance of the Envisat-derived ice motions. In this study, a high correlation is also identified between winter (November-May) SIF estimates derived from the NSIDC and AMAR-E datasets (Figure 2, R=0.87 with results in Kwok (2007), and R=0.93 with estimates in Curry et al. (2014)). On average, the NSIDC-based winter SIF



estimate at passage C (near Davis Strait) is smaller by (-24.3±63.7) ×$10^3$ km$^2$ than those provided by Kwok's, and less than

Curry's estimate by (-45.5±61.0) ×$10^3$ km$^2$. These differences correspond to a few percentages(about 4.5% and 8.9%) in

reference to the average winter SIF estimate based on NSIDC data (511×$10^3$ km$^2$). Overall, a reasonably good agreement is

observed between the NSIDC-derived SIF estimates and earlier results (Figure 2).

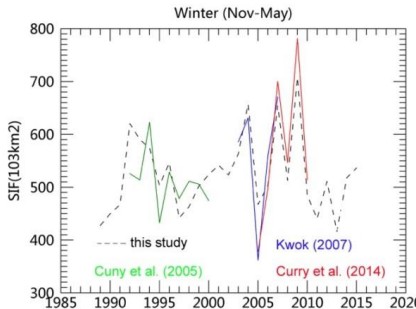

**Figure 2.** Comparison of NSIDC-based SIF estimates with those from earlier studies

**3.2 Spatial variability for ice drift**

**3.2.1 Drift pattern in the Baffin Bay**

The mean monthly sea ice drift pattern in Baffin Bay from 1988-2015 is shown in Figure 3. The prevailing sea ice

10   circulation is southward and primarily confined to the west side of the bay. Most sea ice flows of Baffin Bay originate from

the northern sounds (Smith Sounds, Jones Sounds, and Lascater Sounds). Seasonally, larger SIM is observed during the

winter months (October to May, on average ~10 km/day) than summer months (June to August, on average ~1 km/day).

From October to December, the sea ice coverage in the bay expands very quickly and the SIM accelerates rapidly due to the

stronger wind forcing (Figure 3j-l). Over the late winter and early spring months (January to April), the sea ice speed

15   becomes relatively higher than that in other seasons (Figure 3a-d), especially over the southern Davis Strait region, where the

sea ice attains a speed of 15-20 km/day.



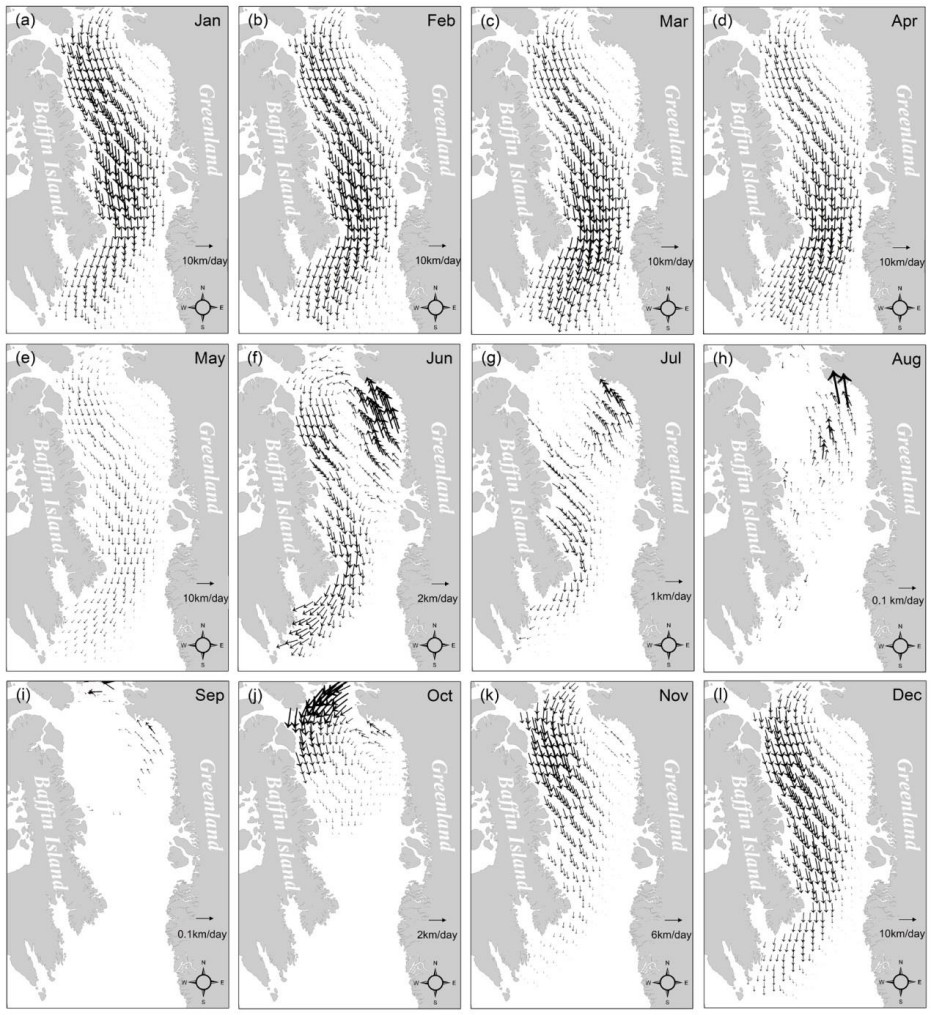

**Figure 3.** Monthly mean sea ice motion fields over the period from 1988-2015 in the Baffin Bay area.

Regionally, a gradient in the sea ice drift speed appears to occur from the east towards the west of the bay. The ice

motion along the west coast of Greenland is comparatively small and even reversed to northward in latitudes north of 70 °N.

5    This northward flowing pattern is linked to the cyclonic ice drift pattern associated with atmospheric and oceanic forcing

(Melling et al., 2001). However, this pattern is not readily visible during the winter months (October-May, Figure 3j-l, a-e).

In contrast, over the summer months (June-August, Figure 3f-h), the ice speed on the east and west sides of the bay show

comparable magnitudes in sea ice speed, thus allowing for the differentiation of the cyclonic sea ice drift pattern.

### 3.2.2 Cross-gate SIF distribution

10   The cross-gate southward components of the annual mean daily SIF (1988-2015) for different passages in Baffin Bay as well

as the FS are given in Figure 4. For comparison, the SIF fields for FS, around 79 °N as used in Kwok (2007), is also



presented. Spatially, the coast grids are restricted to zero flow. The large standard deviation for each plot is indicative of a distinct seasonal and interannual variability in the daily SIF across each passage (Figure 4).

In the north passage (Figure 4a), a rapid increase is observed off the west coast to a peak daily SIF value, which is ~120 km$^2$ at the 6$^{th}$ grid, but smoothly decreases to 92 km$^2$ at the 13$^{th}$ grid near the east coast. Spatially, larger ice motions are skewed towards the west half of this gate. Similar variations are also observed in the south passage (Figure 4c), although with elongated and smoothed distributions across the passage. This larger (smaller) ice speed in the west (east) half of the gate is associated with the cyclonic drift pattern as mentioned above.

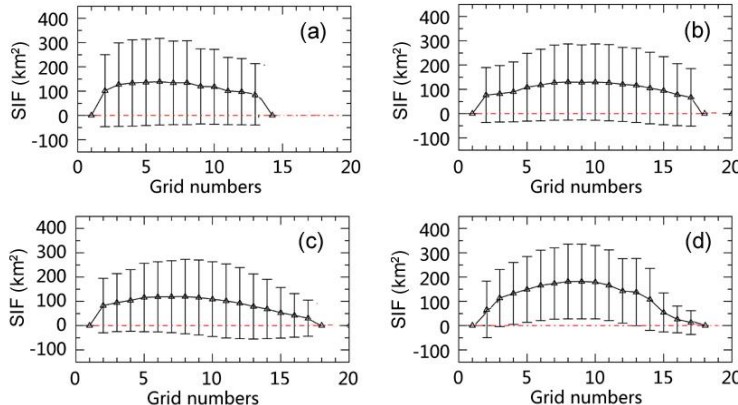

**Figure 4.** Cross-gate distribution of annual (October-September) mean daily SIF over the period from 1988-2015

The distribution in the middle passage (Figure 4b) is different, with the maximum daily SIF observed around the central part (about 105 km$^2$) and the cross-gate distribution appearing to be symmetric. In the FS area (Figure 4d), the SIF distribution plot shows a sharp increase from the west to the central grids (for example, approximately 150 km$^2$ at the 9$^{th}$ grid), and then a steep decrease towards the east end. The daily SIF for most grids at the three passages in Baffin Bay is comparable in the annual average amount, with magnitudes of approximately 100 km$^2$. However, a distinct seasonal variability is observed between the daily SIF for the north and south gates. As indicated in Figure 3, the average SIFs during the winter months (December to May) or autumn (October to November) months for the north passage (Figure 3a) appear to be smaller or larger in comparison with those at the south passage (Figure 3b), respectively.





### 3.3 Variability and/or trend of SIF through different passages

### 3.3.1 Daily SIF variations

The variability of the daily SIF for the period from 1988-2015 is shown in Figure 5 (thin black line). A 30-day smoothed SIF is also added in the plot. A remarkable seasonal variability is found in the daily SIF for each passage in Baffin Bay (Figure

5a-c). Generally, limited daily sea ice export occurs in warm seasons (June-October) while substantially enhanced daily SIF ($\sim 3.0 \times 10^3 \, km^2$) occurs over the cold months (November- May). Moreover, a quasi-decadal change is also demonstrated in Figure 5 (smoothed SIF) between the two time periods 1988-2000 (red) and 2001-2015 (blue).

In Baffin Bay, the seasonal evolution of the daily SIF fields for the three passages is largely similar, but with differences in the details. Commonly, a clear autumn increase (October-December) is observed from the zero outflow for the

three passages, and it is followed by a relatively stable and large winter export (January-May), and then by a rapid spring (since May) or summer (since Jun) decrease toward the zero flux. The increasing slope of the   SIF during the autumn months is sharp and tends to be slower from the north towards the south passages, with a rate of $0.05 \times 10^3 \, km^2$/day, $0.04 \times 10^3$ $km^2$/day, and $0.03 \times 10^3 \, km^2$/day for passage A (Figure 5a), B (Figure 5b), and C (Figure 5c), respectively. In contrast, the springtime decreasing slope of the daily SIF fields is relatively smoother and alike for the three passages, with declining

rates of $-0.021 \times 10^3 \, km^2$/day, $-0.023 \times 10^3 \, km^2$/day, and $-0.024 \times 10^3 \, km^2$/day for passages A , B, and C, respectively.

From late November until May, relatively large daily SIFs together with a slow decrease are observed at the north passage (Figure 5a) from $\sim 3.0 \times 10^3 \, km^2$ to $\sim 2.0 \times 10^3 \, km^2$, which corresponds to a rate of $-0.005 \times 10^3 \, km^2$/day. This trend resembles the declining rate of the daily SIF series ($-0.048 \times 10^3 \, km^2$/day for December to April) in the middle passage (Figure 5b), which also shows a two-mode pattern (one in January and the other in April). In the south passage (Figure 5c),

relatively stable and high daily average ice export ($3.20 \times 10^3 \, km^2$) appear during the months from mid-January to early April. Additionally, two modes seem to exist in the south gates (one in late January and the other in late March). The shape of the daily SIF time series for the middle passage (Figure 5b) manifest as a transitional pattern between those of the north (Figure 5a) and south (Figure 5c) passages. The SIF variability in the FS shows a distinguished pattern compared with the pattern in Baffin Bay, particularly with the nonzero daily SIF values ($\sim 0.5 \times 10^3 \, km^2$) during the summer months (Figure 5d). The high

sea ice export through the FS (on average, $2.6 \times 10^3 \, km^2$/day) is maintained for the cold months from October until May. In addition, two noticeable modes, $2.6 \times 10^3 \, km^2$ (mid-December) and $2.8 \times 10^3 \, km^2$ (mid-March), are reflected in the daily SIF fields of the FS (Figure 5d).

As indicated in Figure 5a and b, decadal alterations in the daily SIF fields are prominent during the cold months in the north and middle passages. Compared with the earlier period (1988-2000), the recent (2001-2015) daily SIF values in

passages A and B (Figure 5a and b, red and blue lines) both increased by approximately $1.0 \times 10^3 \, km^2$ between the time series from December to April. In contrast, the decadal change in the south passages (Figure 5c) is not readily identifiable. During



the warm months such as in the summer (June-July) and autumn (October-November) seasons, the daily SIF was reduced in the recent period in the passages of Baffin Bay ( by $-0.20\times10^3\,\mathrm{km}^2$ (Figure 5a), $-0.28\times10^3\,\mathrm{km}^2$ (Figure 5b), and $-0.23\times10^3$ $\mathrm{km}^2$ (Figure 5a) on average). In the FS area, complex decadal changes in the daily SIF fields are shown in Figure 5d, with a clear short-period increase of $0.5\times10^3\,\mathrm{km}^2$ from October to early December.

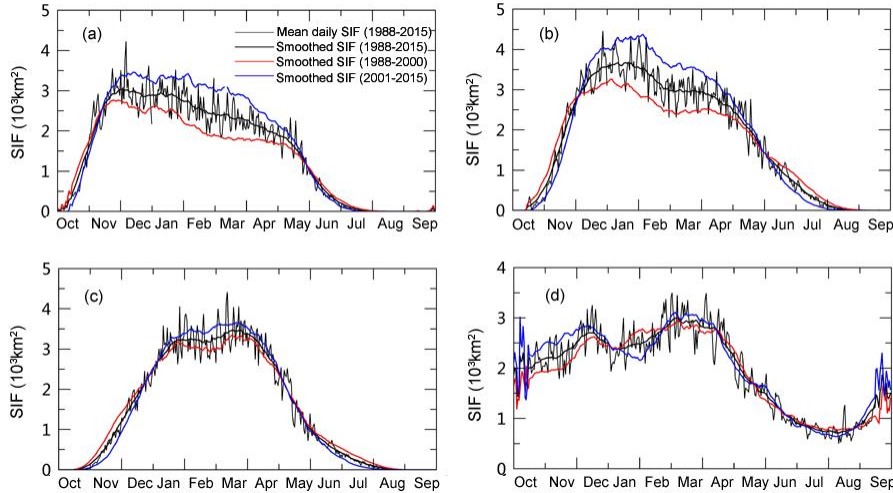

**Figure 5.** Mean daily SIF for different passages over different periods. The SIF through passages A, B, and C inBaffin Bay and the FS are indicated in Figure 5a-d. The thin black line denotes the daily SIF through each passage and the thick black line corresponds to a 30-day smoothed field. The red and blue lines refer to the 30-day smoothed daily SIF fields for the periods prior to (1988-2000) and after 2000 (2001-2015), respectively.

### 3.3.2 Variability and trends in monthly sea ice export

The monthly SIFs through different passages in Baffin Bay and the FS are shown in Figure 6. For the winter months (November to May), the monthly SIF is generally large at the passages in Baffin Bay. The maximum monthly SIF fields for passage A emerges in December ($90\times10^3\,\mathrm{km}^2$), whereas the peak values for B and C occur later, in January ($110\times10^3\,\mathrm{km}^2$) and March ($108\times10^3\,\mathrm{km}^2$), respectively. For the summer months (June to September), the SIFs for Baffin Bay passages are mostly negligible. The SIF through the FS exhibits a steadily large value for the winter months (October to May, roughly $80\times10^3\,\mathrm{km}^2/\mathrm{month}$), and notable ice exports during the summer months (June-September, $\sim30\times10^3\,\mathrm{km}^2/\mathrm{month}$). The large standard deviations, suggested in Figure 6, confirm a significant interannual variation in the monthly SIF for each passage.

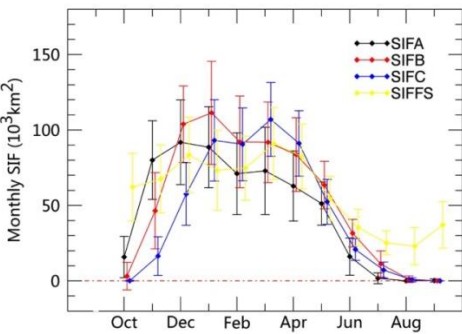

**Figure 6**. Mean monthly SIF for different passages over the period from 1988-2015. Error bars correspond to the standard deviations for the same period.

To further investigate the variability in the monthly SIF at different passages, standardized SIF fields (i.e anomaly map) were calculated and are presented in Figure 7. For passages A and B, the anomaly map is broadly in coincidence (Figure 7a vs b). In particular, anomalous large sea ice exports principally occur during the winter months (November-May) over the period 2003-2011 (Figure 7a and b), which is inconsistent with the lower-than-average monthly SIF during the winter months prior to this time range (1988-2002), thereby providing the preconditions for the observed increasing trend in the wintertime monthly SIF fields for the overall period (1988-2015) (Table 1). Meanwhile, the three passages in Baffin Bay experienced a distinct decline in monthly ice export during the summer months, as ice started to vanish earlier and form later since the turn of the century owing to an increasingly warming climate. Passage C (Figure 7c) and the FS (Figure 7d) do not reveal a distribution pattern with anomaly fields that would favor any significant trend in wintertime monthly SIF. The detailed statiscal results are summarized in Table 1.

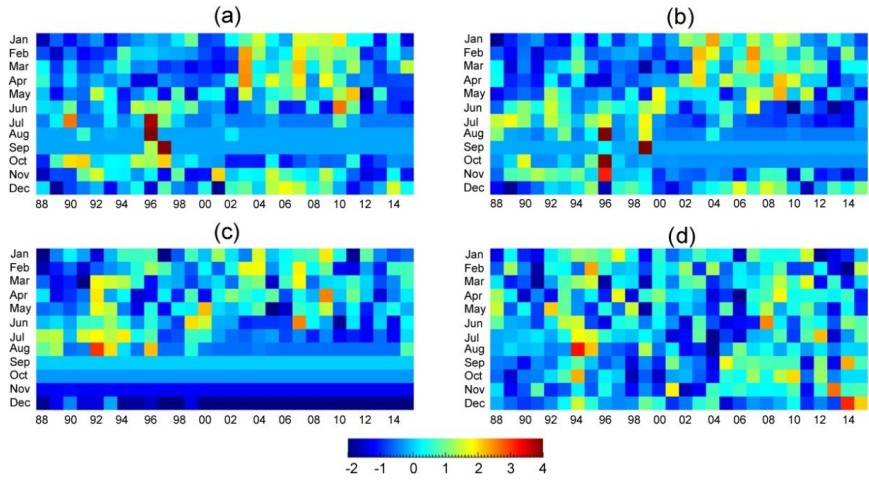



**Figure 7.** Standardized monthly SIF for the period 1988-2015

With respect to trends, the monthly SIF through passage A (Table 1) presents opposite behavior for different seasons. An increased SIF trend predominates over most winter months (Table 1). During the summer months (June-August), the SIF fields, however, basically display a declining trend, although a significance change is not observed (except July). Over this

passage, all months shows a significant decline in SIC, which is especially remarkable during the summer months (Table 1). The reduction in SIC may indicate a trend towards a looser ice pack, which could have two-side effects on the observed SIF trend. Reduced SIC promotes a negative SIF trend through the multiplication effect (Equation 1), and the summertime declining trend of SIF is mostly dictated by the weakened meridional winds caused by a negative trend in the SLPD, and further enhanced by the decline in SIC; and reduced SIC contributes to faster SIM and may promote the SIF. In particular,

the positive SIM trend over the winter months is generally accompanied by a negative SIC trend (Table 1 and Figure 8). The small and statistically insignificant trends in the wintertime cross-gate SLPD (Table 1 and Figure 9) has suggested that geostrophic winds play a secondary role in the increasing trend of wintertime SIM fields. Positive SIM trends in winter largely cancel out the negative multiplication effects related to the reduced SIC, result in an overall increasing SIF trend at that time (Table 1 and Figure 8).

A similar variability in monthly trends is identified for passage B, although the SIF trends are generally greater than those of   A (Table 1). Clearly, the increased SIM during the winter months occurs in concert with a decreased SIC (Figure 8a-e, i). The negative SIF trends continuing from June to November are a combined effect associated with the decreasing trends in both the SIM and SIC fields (Figure 8f-k). Additionally, the SLPD trend contributes a small part to the SIM and SIF trends (Figure 9 and Table 1). For the south passage (C), the monthly SIF trends that reach significant values are

relatively fewer (5 out of 12) and the negative SIF trends persist for a longer period from June to December. Much larger declining SIC trends occur around this passage (Table 1). In the FS, only two months (September and November) present significantly large positive SIF trends (Table 1). The SIM and SIC trends are mostly insignificant and largely cancelled out by each other. Therefore, the significant monthly SIF trends are less common in the FS, which is consistent with the findings provided by Kwok (2009).

Seasonally, the large positive monthly SIF increases are observed in passages A and B for the winter months (October-May) on the average order of $6.78 \times 10^3 km^2/de$ and $7.26 \times 10^3 km^2/de$, respectively (Table 1). In contrast, a slight positive trend occurs at C ($0.52 \times 10^3 km^2/de$). For the summer months (Jun-Sep), the average declining SIF trends over Baffin Bay passages reach to $-1.30 \times 10^3 km^2/de$ (A), $-3.68 \times 10^3 km^2/de$ (B), and $-2.04 \times 10^3 km^2/de$ (C), which is in contrast to the positive one for FS ($1.33 \times 10^3 km^2/de$) (Table 1).






**Table 1.** Monthly and seasonal SIF Trends at THE passages of Baffin Bay and FS over the period 1988-2015. The corresponding trends of SIM, SIC, and SLPG are also given. The different confidence levels are marked with different colors. Red, blue, and gold imply a significance at the 99%, 95%, and 90% levels, respectively.

| | Jan | Feb | Mar | Apr | May | Jun | Jul | Aug | Sep | Oct | Nov | Dec | Winter average (Oct-May) | Summer average (Jun-Sep) |
|---|---|---|---|---|---|---|---|---|---|---|---|---|---|---|
| **Gates A** | | | | | | | | | | | | | | |
| SIF ($10^3$km²/de) | 15.07 | 14.01 | 12.56 | 5.16 | 2.86 | -3.60 | -1.41 | -0.03 | -0.17 | -9.15 | -0.05 | 13.80 | 6.78 | -1.30 |
| SIM (km/de) | 0.99 | 1.23 | 0.98 | 0.42 | 0.36 | -0.29 | -0.16 | 0.01 | -0.02 | -0.72 | -0.14 | 1.08 | 0.53 | -0.12 |
| SIC (%/de) | -0.75 | -0.79 | -0.56 | -0.42 | -5.78 | -13.42 | -6.72 | -1.46 | -1.29 | -13.97 | -2.85 | -1.12 | -3.28 | -5.72 |
| SLPD (Pa/de) | -0.10 | 0.31 | 0.00 | 0.29 | 0.26 | -0.70 | -0.62 | -0.32 | 0.51 | 0.24 | 0.60 | 0.19 | 0.22 | -0.28 |
| **Gates B** | | | | | | | | | | | | | | |
| SIF ($10^3$km²/de) | 20.59 | 17.49 | 12.90 | 10.29 | 3.89 | -5.83 | -7.91 | -0.95 | -0.01 | -3.48 | -16.38 | 12.81 | 7.26 | -3.68 |
| SIM (km/de) | 1.25 | 1.30 | 0.85 | 0.71 | 0.31 | -0.34 | -0.72 | -0.15 | -0.002 | -0.30 | -1.12 | 0.93 | 0.49 | -0.30 |
| SIC (%/de) | -2.63 | -0.98 | -0.29 | -0.47 | -3.06 | -9.35 | -13.07 | -2.67 | -0.13 | -4.43 | -11.69 | -6.24 | -3.72 | -6.31 |
| SLPD (Pa/de) | 0.07 | 0.40 | -0.17 | 0.30 | 0.04 | -0.85 | -0.40 | -0.52 | 0.38 | 0.09 | 0.01 | 0.36 | 0.14 | -0.35 |
| **Gates C** | | | | | | | | | | | | | | |
| SIF ($10^3$km²/de) | 6.30 | 8.42 | 4.72 | 4.98 | 0.10 | -2.49 | -4.92 | -0.73 | 0.00 | -0.15 | -10.32 | -9.86 | 0.52 | -2.04 |
| SIM (km/de) | 0.48 | 0.75 | 0.47 | 0.43 | 0.05 | -0.10 | -0.44 | -0.11 | 0.00 | -0.02 | -0.80 | -0.57 | 0.10 | -0.16 |
| SIC (%/de) | -9.67 | -6.94 | -4.56 | -4.03 | -5.74 | -7.00 | -9.68 | -1.52 | 0.00 | -0.43 | -9.38 | -9.37 | -6.27 | -4.55 |
| SLPD (Pa/de) | -0.18 | 0.22 | -0.62 | 0.62 | 0.00 | -0.76 | -0.38 | -0.24 | 0.22 | 0.22 | -0.28 | -0.06 | -0.01 | -0.29 |
| **Gates FS** | | | | | | | | | | | | | | |
| SIF ($10^3$km²/de) | -2.60 | -3.06 | 5.56 | 2.64 | -3.10 | 0.42 | -0.92 | -1.63 | 7.46 | 8.18 | 11.96 | 6.89 | 3.31 | 1.33 |
| SIM (km/de) | -0.42 | -0.26 | 0.43 | 0.17 | -0.26 | 0.05 | -0.04 | -0.22 | 0.44 | 0.57 | 0.96 | 0.63 | 0.23 | 0.06 |
| SIC (%/de) | -0.93 | -3.10 | -1.96 | 0.13 | -1.43 | -0.58 | -3.26 | 3.83 | 2.35 | -2.27 | -1.94 | -3.69 | -1.90 | 0.59 |
| SLPD (Pa/de) | -0.90 | 0.15 | 0.88 | 0.38 | 0.11 | 1.00 | 0.95 | 1.22 | 0.50 | 0.51 | 1.03 | 0.10 | 0.28 | 0.92 |





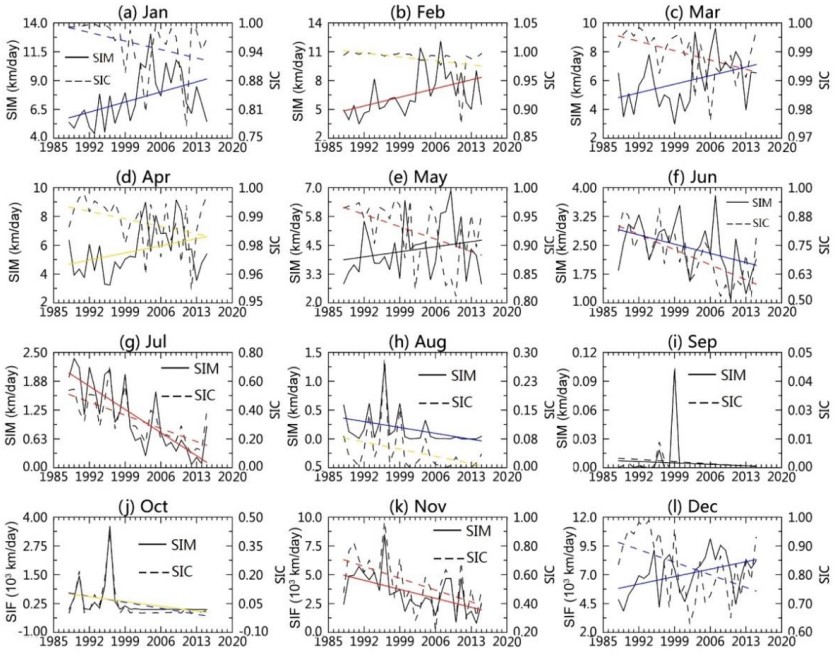

**Figure 8.** Monthly mean SIM (bold line, unit: km/day) and SIC (dash line) at passage B. The linearly fitted line is added, and red, blue, and gold indicate a significance level reaching to 99%, 95%, and 90%, respectively.

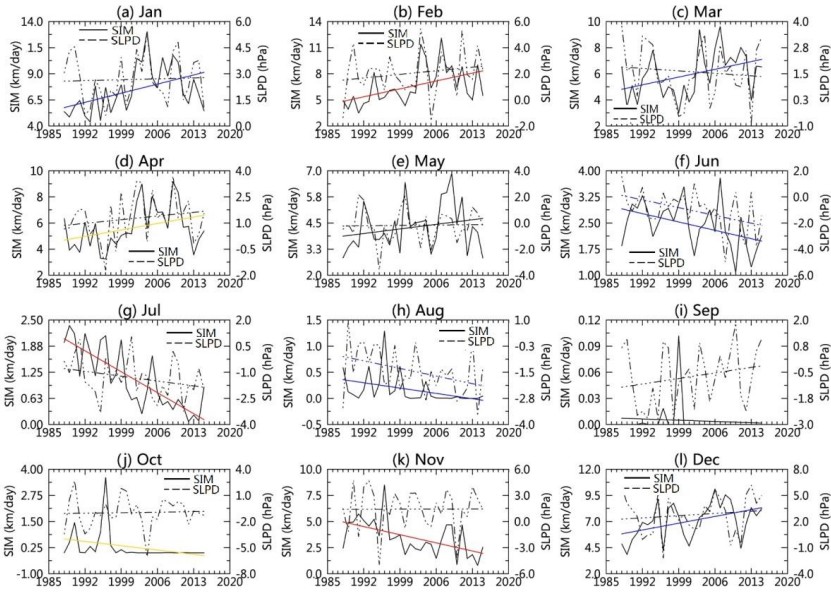




**Figure 9.** Monthly mean SIM (bold line, unit: km/day) together with cross-gate sea level pressure difference (SLPD, dash

line) at B. The linearly fitted line is shown, and red, blue, and gold indicate for a significance at the 99%, 95%, 90% levels,

respectively.

### 3.3.3 Variability and trends for annual and seasonal exports

5   Figure 10 shows the time series of the annual (October to the following September) SIF over the period

1988/1989-2014/2015. The mean annual SIFs for the three Baffin Bay passages over this period are $555 \times 10^3 km^2$ (A),

$642 \times 10^3 km^2$ (B), and $541 \times 10^3 km^2$ (C). These amounts are less than that of the FS ($707 \times 10^3 km^2$). A decadal increase of the

annual SIF is evident for passages A and B (Figure 10a and e) between the two periods: 1988-2000 and 2001-2015. For

passage A, the mean annual SIFs increased by $127 km^2$ for the period before and after 2000, from $489 \times 10^3 km^2$ to

10  $616 \times 10^3 km^2$ (Table 2), respectively, and for passage B, the mean annual SIFs increased by $94 \times 10^3 km^2$ for the period before

and after 2000 from $592 \times 10^3 km^2$ to $686 \times 10^3 km^2$, respectively. Indeed, such clear quasi-decadal changes leads to an overall

increasing trend, as illustrated in the annual SIF series for the two passages (Figure 10a and e). Specifically, passages A and

B present a significant trend of $53.1 \times 10^3 km^2/de$ and $41.2 \times 10^3 km^2/de$, respectively, whereas the annual SIF trends at passages

C and the FS present unclear trends (Figure 10i and m) and decadal changes that are not prominent (Table 2).

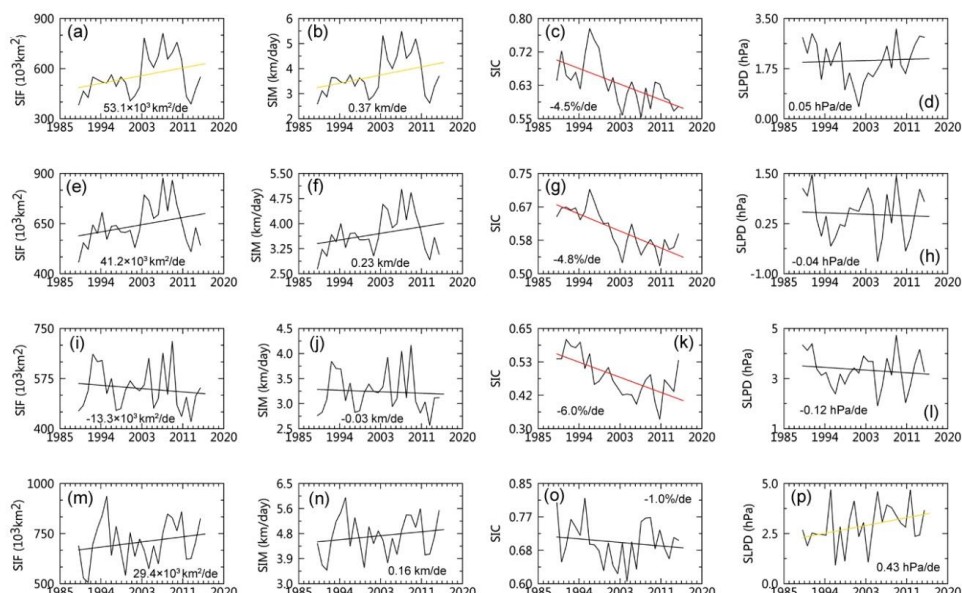

**Figure 10.** Time series for the annual SIF (October to following September) through different passages. The x-axis label

denotes the year of September. For example, year '1994' indicates the sea ice seasonal cycle of 1993/94. The annual mean





SIM, SIC, and SLPD fields are also given in the second, third, and fourth columns, respectively. The Linearly fitted line is superimposed with red, blue, and gold colors, which indicate significance levels of 99%, 95%, 90%, respectively.

For passages A and B, the increasing SIM trend (Figure 10b and f) is primarily caused by a positive SIF trend, which is expected to be associated with the markedly decreasing SIC (Figure 10c and g) as mentioned above. Again, negligible SLPD

trends appear to play a minor role (Figure 10d and h). A slight negative SIF trend is found in passage C (Figure 10i). Compared with the three passages in Baffin Bay, the FS shows a relatively significant annual mean SPLD trend on the order of 0.43 hPa/de at the confidence level of 90% (Figure 10p), thus, this parameter plays an important role in increasing the SIM (0.16 km/de) and SIF ($29.4 \times 10^3 km^2$/de) trends. With respect to the SIC fields, the declining trends for the three Baffin Bay passages (from -4.5%/de to -6.0%/de) are dramatic in relation to that of the FS (-1.0%/de). Therefore, SIC changes in

the Baffin Bay area may explain more fractions because of the relatively significant increasing SIM trends compared with that of the FS region.

The annual SIF variability and trends  are primarily determined by winter sea ice export from October to next May. With regard to the three Baffin Bay passages, no less than 93% on average of the annual SIF is attributable to winter export for the period 1988/1989-2014/2015 (Table 2 and Figure 11). By comparison, this value is comparatively lower for the FS at

87%. Moreover, the average contribution from winter export is augmented by 2%~6% (Figure 11), depending on the passages, for the later period (2001-2015) in comparison with the earlier period (1988-2000). However, a decline of approximately 2% is occurs for the winter contribution of sea ice exported through the FS (Figure 11). Based on the annual ice export balance, the elevated percentage of winter sea ice export means that  the contribution from the summer export is abated, and vice versa.

**Table 2.** Mean annual and seasonal SIF (unit: $10^3 km^2$) for different periods. The winter and summer estimates refer to the temporal spans of October-May and June-September, respectively. Numbers in a form of 'N1/N2/N3', as listed in Table 2, is referred to the annual, winter, and summer SIF fields, respectively.

| Passages | 1988-2000 | 2000-2015 | 1988-2015 |
|----------|-----------|-----------|-----------|
| A | 489/464/25 | 616/603/13 | 555/535/20 |
| B | 592/534/58 | 686/656/30 | 642/597/45 |
| C | 545/509/46 | 536/514/22 | 541/511/30 |
| FS | 692/610/82 | 720/624/96 | 707/616/91 |





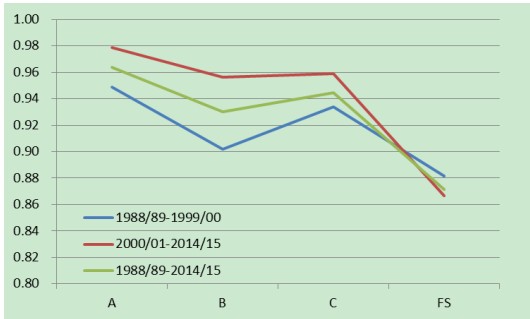

**Figure 11.** Fraction of winter sea ice export relative to the annual SIF for different periods

### 3.3.4 Net sea ice transport between different passages in Baffin Bay

The net sea ice transport is an indicator of the ability to reserve sea ice for a regime, and it is estimated by taking as as the

SIF difference (SIFD) between two passages. For instance, SIFD_AB corresponds to the SIFD between passages A and B

(i.e. A-B). A positive value suggests a net ice inflow (convergence), whereas a negative value refers to a net outflow

(divergence). Table 3 lists the statistical results for the net sea ice transport between different passages in Baffin Bay over the

period from 1988/1989-2014/2015. Annually, a distinct net outflow ($-87.3 \times 10^3$ km$^2$/a on average) in the north regime of

Baffin Bay (the areas between passages A and B, SIFD_AB). That is, more sea ice is exported across passage B, compared

with passage A. For SIFD_AC, the regime between passages A and C, approaches a nearly balanced state in terms of sea ice

inflow and outflow, with a small amount of net input of $14.3 \times 10^3$ km$^2$/a (Table 3). In addition, a much larger mean value of

annual SLPD_BC, on the order of $101.6 \times 10^3$ km$^2$/a, is suggestive of a sea ice reservoir within the southern part of Baffin

Bay (between passages B and C).

**Table 3.** Statistical results for the net annual SIF (1988/1989-2014/2015) between different passages in Baffin Bay. The red

and blue colors denote a trend that greater than than the significance levels of 99% and 95%, respectively.

|  | SIFD_AB | SIFD_BC | SIFD_AC |
|---|---|---|---|
| Mean ($10^3$ km$^2$) | -87.3 | 101.6 | 14.3 |
| Standard Deviation ($10^3$ km$^2$) | 57.7 | 87.8 | 118.6 |
| Trend ($10^3$ km$^2$/de) | 11.9 | **64.2** | **54.5** |

Seasonal fluctuations for the net ice transport between different passages are also obvious (Figure 12). For SLPD_AB,

negative values greater than $-25 \times 10^3$ km$^2$ (i.e. net sea ice outflow) occur for most months, except for January and February

(Figure 12, black line) when a net positive sea ice inflow of less than $40 \times 10^3$ km$^2$ is observed. For SLPD_AC, positive

quantities smaller than $70 \times 10^3$ km$^2$ (net sea ice inflow) are identified from January to March, whereas negative values (i.e.,




net outflows) not exceeding $-35 \times 10^3$ km$^2$ are found for the remaining months (Figure 12, blue line). Regarding SLPD_BC, larger positive amounts (lower than $50 \times 10^3$ km$^2$/month) are found over the later winter period from February until April, whereas smaller positive values (below $10 \times 10^3$ km$^2$/month) are encountered from August to November (Figure 12, red line). The negative values in SLPD_BC greater than $-15 \times 10^3$ km$^2$/month occur during the mid-summer months (June to July), thus

implying a much reduced input via passage B. As a reservoir pool of sea ice (as mentioned above), the net positive sea ice transport between passages B and C (SLPD_BC) is largely dictated by those from February to April (on average, approximately up to $40 \times 10^3$ km$^2$/month) (Figure 12, red line).

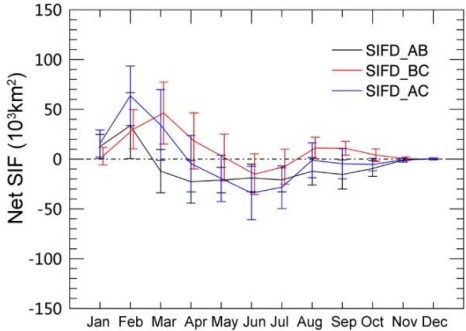

**Figure 12.** Mean net monthly SIF between different passages in Baffin Bay over the period from 1988/1989-2014/2015. The

Error bar corresponds to the standard deviation.

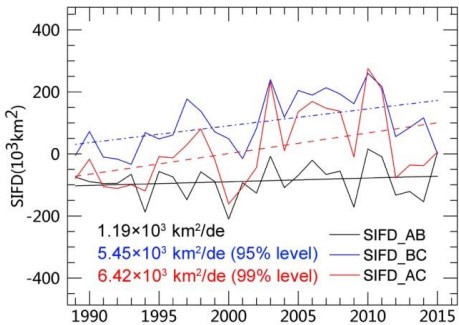

**Figure 13.** Net annual SIFD between different passages. For example, SIFD_AB is the difference between SIF through passages A and B. Linearly fitted line is also added, with dot line, dash line, and dash dot line representing trends reaching the 90%, 95%, and 99% confidence levels, respectively.

We further analyzed the net annual SIF trend (Figure 13). SIFD_AB presents a clear trend ($11.9 \times 10^3$ km$^2$/de), whereas the other two fields reveal significant positive trends of $54.5 \times 10^3$ km$^2$/de (SIFD_AC) and $64.2 \times 10^3$ km$^2$/de (SIFD_BC),





suggesting that Baffin Bay, especially the southern part between passages B and C, tends to present more converged sea ice over time. With respect to SIFD_AC and SIFD_BC, the increasing trends are mainly associated with the larger winter net ice export from December to May (Table 4). Particularly, the greatest net ice flux approaching $23 \times 10^3$ km$^2$/de occurs in December (Table 4), which is related to the increasing SIF trend in in A and B by up to $13 \times 10^3$ km$^2$/de together with a

decreasing SIF trend in C by up to ~ $-9 \times 10^3$ km$^2$/de (Table 1). Seasonally, an accelerated trend for ice inflow through A or B during the winter and spring months (from January to May) is consistent with a relatively slower outflow via C (Table 1), which further implies that during the cold seasons sea ice within the south part of Baffin Bay is likely converging to a higher degree. During the summer months (June-August), a positive SLPD_AB trend accompanies a negative SLPD_BC trend . As summarized in Table 1, these trends are mainly caused by a faster decline of the SIF through passage B during the summer

months ($-3.68 \times 10^3$ km$^2$/de), compared with the other two passages ($-1.30 \times 10^3$ km$^2$/de for A and $-2.04 \times 10^3$ km$^2$/de for B).

**Table 4.** Trends for the monthly SIF difference through different gates. Red, blue, and gold colors denote a trend that approaches the significance levels of 99%, 95% , and 90%, respectively.

| | Jan | Feb | Mar | Apr | May | Jun | Jul | Aug | Sep | Oct | Nov | Dec |
|---|---|---|---|---|---|---|---|---|---|---|---|---|
| SIF_AB ($10^3$km$^2$/de) | -5.51 | -3.47 | -0.34 | -5.12 | -1.03 | 2.23 | 6.49 | 0.92 | -0.16 | -5.67 | 16.43 | 0.99 |
| SIF_BC ($10^3$km$^2$/de) | 14.29 | 9.06 | 8.18 | 5.31 | 3.79 | -3.34 | -2.98 | -0.22 | 0.01 | -3.33 | -6.06 | 22.66 |
| SIF_AC ($10^3$km$^2$/de) | 8.78 | 5.59 | 7.84 | 0.19 | 2.76 | -1.12 | 3.51 | 0.70 | -0.17 | -9.00 | 10.36 | 23.66 |

## 4. Discussion

### 4.1 Linkages to changes in climate factors

Abrupt decadal changes in the annual (Figure 10a and e) and winter SIF fields (Table 2) are apparent for the north and middle passages of Baffin Bay. In this section, we further investigate four climatic factors, including the SW,SLP, SAT, and SST, which may contribute to these changes. All of the used climatic variables are NOAA NECP/NCAR reanalysis products (Kalnay et al., 1996).

   The changes of the local SW between two periods, defined as P1(1988-2000) and P2 (2001-2015), vary with season

(Figure 14a and b). For the winter period (October-May), the inter-period changes (P2-P1) show that the southeastward SWs (with magnitudes mostly less than 0.4m/s) dominate over passages A and B (Figure 14a), which is consistent with the increasing trend of the SIF as observed during most winter months (Table 1) as well as the enhanced annual and winter export for the two passages (Table 2). In addition, the anomalous easterly SW (0.3~0.4 m/s) to the south of Davis Strait would push sea ice against Baffin Island in the west (Figure 14a); hence, it may have exerted a subtle influence on boosting a

southward sea ice advection through the southern passage C. As a result, the average trend of the wintertime monthly SIF



between P1 and P2 (roughly $0.51 \times 10^3$ km$^2$/de, as shown in Table 2), and the decadal change of winter (October-May) sea ice export (by approximately $5 \times 10^3$ km$^2$ (Table 3), from $509 \times 10^3$ km$^2$ (P1) to $514 \times 10^3$ km$^2$ (P2), are not readily evident for passage C. During the summer seasons (Figure 14b), the inter-period changes of the SW fields in Baffin Bay mainly present a northwestward direction, with increasing magnitudes from southern to northern Baffin Bay (as indicated by background

color in Figure 14b). To be specific, the average declining trend in the SW magnitude is as follows:    1.1 m/s for passage A , 0.8m/s for passage B, and 0.6 m/s for passage C (Figure 14b). This decline is associated with the negative summer SIF trend as observed for each passage in Baffin Bay (Table 1, Jun-August).

The spatial layout in the SLP, on the other hand, sheds light on the role of the geostrophic winds in the modulation of sea ice transport through Baffin Bay (Figure 14c and d). For the winter period, the inter-period SLP alternations, with higher

SLPs (on average, approximately 0.9 hPa) in west Baffin Island and relatively lower SLPs (approximately 0.6 hPa) spanning over Baffin Bay (Figure 14c), seem to support a northwesterly wind movement through passages A and B. Lower SLPs (approximately 0.4 hPa) are observed in close proximity to the southern passage (C) as well as in northern Labrador Sea (Figure 14c). This SLP distribution pattern has the potential to promote a westward air movement, which is broadly consistent with the local SW as shown in Figure 14b. In summer, the higher SLPs (approximately 2.5 hPa) over the east side

of Baffin Bay (Greenland coasts) along with the lower SLP (about 0.5 hPa) over the west side (Figure 14d) are favorable for a northwestward migration for the air mass atop the Baffin Bay area (Figure 14b). Broadly, the changes of SLP distribution are consistent with the spatial variations of S,W, which is largely in phase with the spatiotemporal changes of sea ice export through different passages over Baffin Bay.




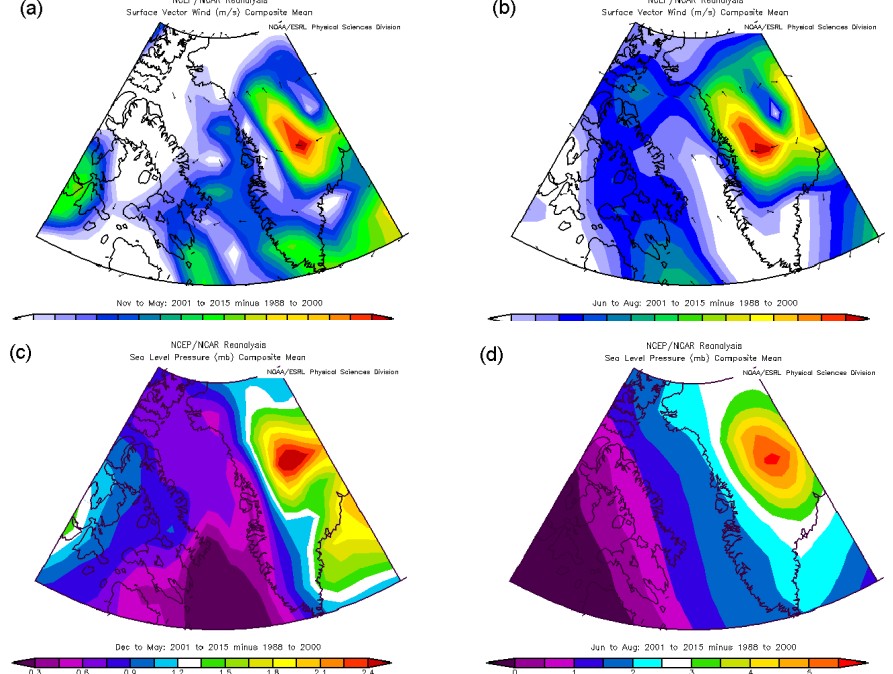

**Figure 14.** Inter-period changes (P2-P1) of SW for the (a) winter (November-May) and (b) summer (June-August) between

P1(1988-2000) and P2 (2001-2015). The corresponding SLP changes for the (c) winter and (d) summer are also shown.

Furthermore, we examined the changes of two climatic factors (SAT and SST) to interpret the Baffin Bay sea ice export

variations in the context of an amplified warming climate in the Arctic (Serreze et al., 2009; Screen et al., 2013). Between

the two periods (P2-P1), the SAT and SST are reinforced for certian seasons (Figure 15). The warmer surface air and upper

ocean are consistent with facts reported by Zweng and Münchow (2006), and generally congruent with the notable reduction

in SIC fields as highlighted in Figure 10 and Table 1.

During the winter months, the declining trend in the SIC over different passages is significant (Table 1) and consistent

with a warmer SAT (Figure 15a) and SST (Figure 14b) throughout the bay. Particularly, the warmest winter SAT (mostly

above 3.5 ℃, Figure 15a) and SST (dominantly beyond 4.0 K, Figure 15c) appear in the southeastern part of Baffin Bay,

covering the major portion of passage C. Accordingly, a larger winter SIC reduction (-6.27%/de) is observed especially over

this passage. The enhanced SAT and SST may also play a key role in the significant decline in SIC at passages A (-3.27%/de)

and B (-3.72%/de).




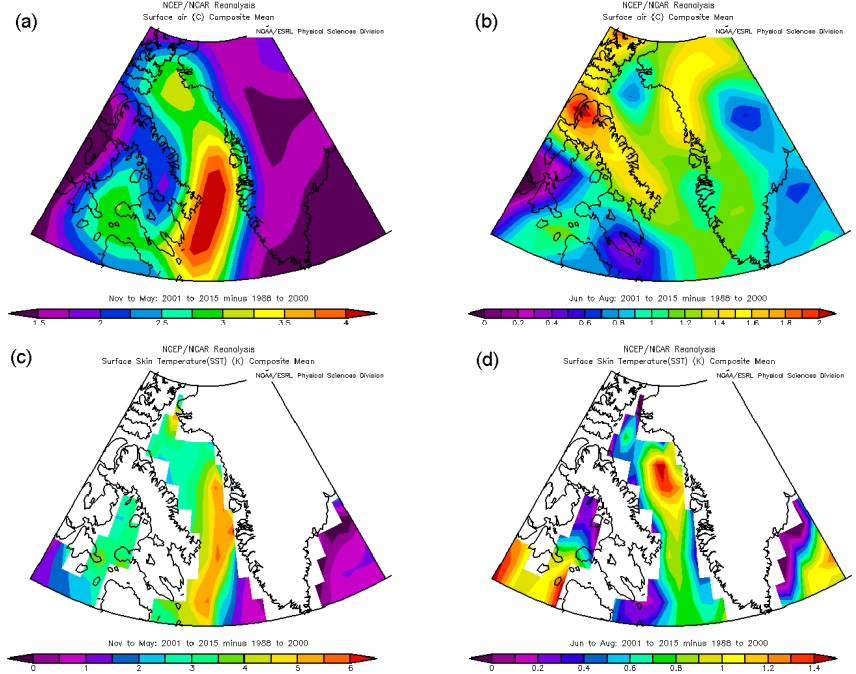

**Figure 15.** Inter-period changes (P2-P1) of SAT between P1 (1988-2000) and P2 (2001-2015) for the (a) winter (November-May) and (b) summer (June-August). The SST changes are presented for the (c) winter and (d) summer seasons.

Over the summer period, the temporal changes of the SAT and SST are more prominent over the central part of Baffin Bay (Figure 15b and d), which is consistent with the largest average decline of the SIC at passage B (-6.31%/de) during summer (Table 1). The SAT and SST increases are relatively weak in passage A (0.8 ℃ and 0.6 K, on average) compared to those of passage C (0.9 ℃ and 1.1 K). However, a slightly faster reducing SIC rate is found at A (-5.72 %/de) than C (-4.55 %/de), which can be explained by the continued occurrence of a large polynya (the North Water Polynya) in the north of the bay that allows for the exposure of abundant open water to trigger the well-documented positive ice-albedo feedback (Melling et al., 2001). More dark water will absorb greater amounts of solar energy that melts more sea ice, such that a swifter summer reduction in SIC is expected at passage A than C. Overall, the remarkable SIC decline, along with the winds blowing northwestward (Figure 14b) appears to have an important influences on the summer negative SIF trends for each passage in Baffin Bay: $-1.3 \times 10^3$ km$^2$/de at A, $-3.68 \times 10^3$ km$^2$/de at B, $-2.04 \times 10^3$ km$^2$/de at C (Table 1).

**4.2 Connections to the NAO and SLPD**

The NAO represents the dominant mode of atmospheric variability over the northern North Atlantic Ocean, and it is closely related to the midlatitude Azores high and the sub-polar Icelandic low pressure systems . For the positive/negative NAO




phase, the SLP is deeper/shallower over the Icelandic low, the atmospheric circulation becomes stronger/weaker, and northerly/southerly wind likely flows through Baffin Bay (Häkkinen and Cavalieri, 2005) and the FS (Kwok, 2000; Kwok et al., 2004; Kwok, 2009). These changes could in part account for the variability of sea ice extent through the passages in Baffin Bay and Laborador Sea (Gunnar et al., 2004). However, Figure 16 suggests that the monthly SIF is only slightly

correlated with the NAO index for the three passages through Baffin Bay (R = 0.23~0.32). Further, only a weak connection occurs between the monthly SIF through the FS and NAO over the period 1988-2015 (R =0.15), which does not mean NAO has a trivial role in modulating the interannual variability of sea ice drift and export anywhere in the Arctic Ocean outlets (Kwok et al., 2013). Previous studies have reported a strong temporal sensitivity of the FS sea ice export to the NAO index, such as for the periods 1979-1996 (R=0.66) and 1979-2007 (R=0.60) (Kwok, 2009). This temporallysensitive association

with the NAO also occurs for the Baffin Bay passages for the two periods (1988–1996, 1988–2015): A (R=0.41, 0.22), B (R=0.45, 0.24), and C (R=0.49,0.30).

The SLPD across a passage represents for the local atmospheric forcing and is closely related to the monthly SIFs through a fluxgate (Figure 17). However, the correlation at Baffin Bay (R=0.69~0.71) is marginally weaker than that for the FS (R=0.74). Baffin Bay is confined by land in the west (Baffin Island) and east (Greenland); hence, the sea ice drift pattern

in Baffin Bay may be readily subject to an orographic configuration. Nonetheless, the dynamic effect of sea ice over the FS area is relatively less susceptible to the small Svalbard island to the east (Kwok et al., 2004), which means a slightly higher degree of free sea ice drift and thus a relatively stronger connection of ice export with local atmospheric circulation in the FS area.

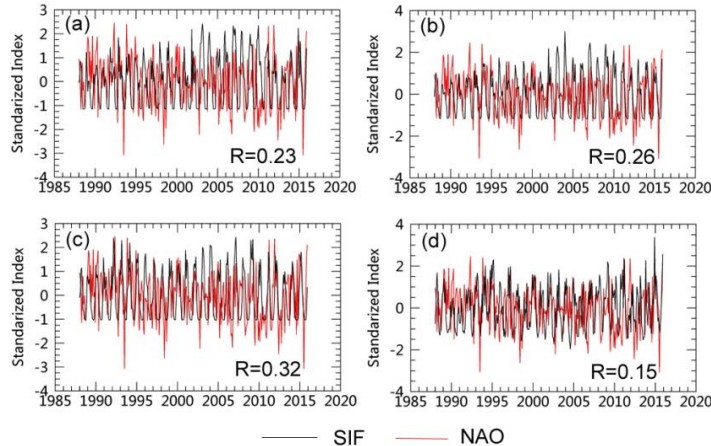

**Figure 16.** Time series of the monthly SIF through passages (a) A, (b) B, and (c) C, and (d) the FS, along with the

corresponding monthly mean NAO index. To facilitate our analysis, the two variables have been standardized. To be specific,





monthly SIF values (or monthly mean NAO values) are first subtracted from the mean monthly value over the period

1988-2015, and then divided by the standard deviation of the corresponding monthly values.

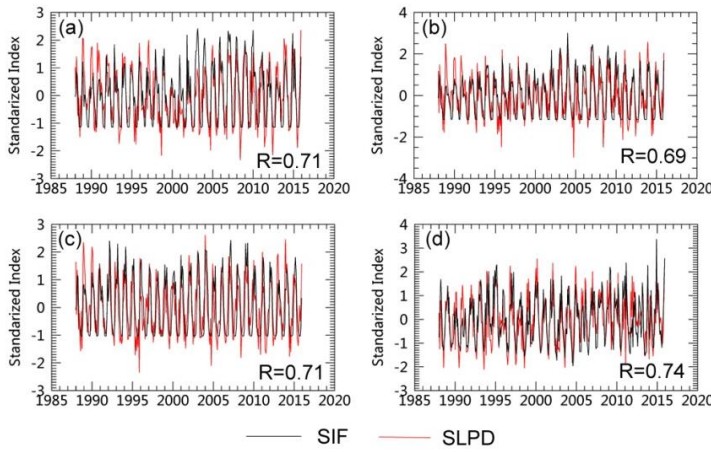

**Figure 17.** Time series of monthly SIF via different passages (including (a) A, (b) B, and (c) C, and (d) the FS) and monthly

mean SLPD. The two parameters are also standardized as depicted in Figure 16.

**5. Conclusions**

The satellite-derived sea ice area exported through three passages within Baffin Bay was obtained over the period 1988-2015.

A comparison shows that our SIF estimates are reasonable consistent with previous results. For the Baffin Bay passages, the

trends and changes in SIF fields are spatiotemporally varying. Seasonally, a general increasing (decreasing) trend in the SIF

is observed over the winter (summer) months. Regionally, an increasing trend of the annual SIF is identified for the north

and middle passages (A and B), whereas an insignificant negative trend is found for the south passage (C). The obvious

increasing trend in the SIF through A and B together with a slight trend through C combined to produce a positive net SIF

trend between passages A and C or between B and C (i.e., net inflow). This suggests a tendency to retain (converge) more

sea ice within the Baffin Bay regime.

The changes and variability of SIF for each Baffin Bay passage are primarily determined by the variations inSIM

associated with fluctuations in the SW and SLP and further modulated by the SIC, which exhibits a significant decline at all

months that is linked to warmer SAT and SST values. The reduced SIC may have made a larg contribution to the increasing

trend of SIM as observed for passages A and B (Figure 10). Compared with the FS (Kwok, 2009), the cross-gate SLPD for

all the passages of Baffin Bay does not present a significant trend and thus does not enhance the trend found    in the SIM or

SIF.



Over the investigated period (1988-2015), the monthly sea ice export through the Baffin Bay passages is overall weakly connected to the atmospheric circulation pattern related to the NAO. Similar to sea ice outflow through the FS (Kwok, 2000; Kwok et al., 2004; Kwok et al., 2013), the association with the NAO is sensitive to the time period examined. However, a relatively robust connection is observed between the monthly sea ice export and the SLPD (R=~0.70 for Baffin Bay

passages). Considering that Baffin Bay is largely constrained by land in the west (Baffin Island) and east (Greenland), it is unsurprising the sea ice drift in Baffin Bay may not be as free as that in the FS (R=0.74).

**Acknowledgement**

The authors would like to thank the data providers as follows. NSIDC provides the satellite-derived ice motion and concentration data; National Centers for Environmental Prediction/National Center for Atmospheric Research

(NCEP/NCAR) provides the reanalysis product. We are grateful for valuable suggestions from J. Zhang, M. Steel, A. Schweiger (University of Washington). This work was supported by the National Natural Science Foundation of China under Grant 41406215, 41706194; the fund provided by the Qingdao National Laboratory for Marine Science and Technology; a NSFC-Shandong Joint Fund for Marine Science Research Centers (U1606401).

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
