# Peer review of "Baffin Bay sea ice inflow and outflow:1978/1979-2016/2017"

_The Cryosphere, 2018_

## Referee Comment (RC1) · Anonymous Referee #1 · 14 Aug 2018

This paper tries to monitor the sea ice flux (SIC) changes in Baffin Bay from 1988 to 2015 using the NSIDC sea ice concentration (SIC) and sea ice motion (SIM) datasets. The authors also try and link these changes in SIF to climate variables from the NCEP-NCARR dataset. Thought the datasets used for this study are very relevant and results can be very relevant to understanding the climate variability of sea ice conditions in Baffin Bay, after reading the paper, I do not think the authors understand the sea ice conditions in Baffin Bay and the drivers of the sea ice fluxes properly.

My first concern is why the authors keep comparing the SIF of Baffin Bay with the SIF of Fram Strait. The ice regimes of those two regions are very different. The source of the SIF from Fram Strait comes directly from the Arctic Ocean and a large portion of the ice that flows there is comprised of multi-year ice all year round. In the case of

[Figure]

Baffin Bay, the ice flows from Lancaster Sound and Nares strait in the summer months and is a mix of multi-year ice and first-year ice but in the winter, the ice mainly comes from first-year ice generated in polynyas of Lancaster Sound and the northern part of Baffin Bay. A major driver in Baffin Bay is the North Water Polynya (NOW) and nowhere do the authors mention this. Also, that polynya exists because of the ice bridge that is created in Nares Strait in the winter which blocks the inflow of thicker multi-year ice from the Arctic Ocean. Fram Strait and Baffin Bay are thus not comparable in terms of SIF.

Another issue is the separation between before and after the year 2000. The selection of the year 2000 seems arbitrary to me. Many climate change studies (including the IPCC report) state that there was a climate shift around 1998. Why did the authors not look at before and after 1998? Why don't there results show this climate regime shift (it should appear in the NCEP-NCARR dataset).

Also, some results can be explained by the sea ice conditions themselves. For example, the low SIM values on the Greenland coast in the winter months can be explained by the ice conditions. The sea ice along the coast is land fast, i.e. attached to the Greenland coast and does not move (note: the extent varies each year). I would strongly recommend that the authors visit the Canadian Ice Service website (https://www.canada.ca/en/environment-climate-change/services/ice-forecasts-observations/latest-conditions.html) and specifically their 30-year ice atlas (https://www.canada.ca/en/environment-climate-change/services/ice-forecasts-observations/latest-conditions/climatology/30-year-climatic-atlases.html) to better interpret the SIF results.

Overall, I would not reject this paper since it is very relevant to the studied field of climate change but I suggest major reviews after the authors better describe the region of interest and its different drivers. I would suggest to remove the comparison with Fram Strait as they don't have the same ice regimes. I would also study a bit more in detail the ice conditions which can be obtained on a weekly basis from the Canadian

Ice Service in order to improve the interpretation of the results.

Specific comments: In the figure caption, I suggest adding more detailed descriptions of the figures. Specify what the a), b), c) and d) subfigures are etc.

For Figure 7, what is the reference to generate the anomaly map? Usually it's a specific period that is used but it was not specified in the text. Also, what are the units if any for this anomaly map?
* * *

---

## Referee Comment (RC2) · Anonymous Referee #2 · 28 Aug 2018

General Comments:

In reference to the review criteria provided by The Cryosphere (https://www.the-cryosphere.net/peer_review/review_criteria.html), I submit the following general commentary on this work.

The originality (novelty) of this work is Fair. It does not represent substantial progress beyond the current scientific understanding of sea ice flux through Baffin Bay or any other area of the Arctic Ocean.

The scientific quality of this work is Fair. The purpose of the work is articulated, however the objectives laid out in the second-to-last paragraph of the introduction section do not strike this reviewer as testable hypotheses. They are rather statements resulting

in the paper becoming something of a data report. The analyses presented are somewhat perfunctory, and potentially these points indicate a lack of understanding of the physical atmospheric and/or oceanic processes acting on the surface (ocean or sea ice) within Baffin Bay. The first mention of the hemispherically significant North Water Polynya does not occur until Line 6 on page 23 of this work. In this vein, it's not really explained that the sea ice in Baffin Bay may be as a result of import from the Lincoln Sea as a result of motion through Nares Strait, Kane Basin, and then Smith Sound before it might either enter Baffin Bay, or encounter an ice-bridge blocking inflow to Baffin Bay from the north. The annual and seasonal presence/absence of this very important ice-bridge feature and the amount of MYI imported from the North is not explicitly investigated in this work. First-year stages of development of sea ice in Baffin Bay may then also have been grown in place within the North Water and exported southward, depending on the formation of the ice-bridge, and the amount of ice in the Bay, which has been imported from the Canadian Arctic Archipelago (CAA), mostly through Lancaster Sound. Since the authors chose a position for their northernmost 'Gate A' south of Lancaster Sound, the flux of sea ice from the CAA in the west, or from Smith Sound in the north, cannot be differentially discerned. Other important considerations that occur to this reviewer include sea ice melt during transport southward within Baffin Bay, especially considering that new and young stages of development (<30cm thick) may be grown and exported from the North Water Polynya. Finally, it might be that the authors have not accounted for the presence of fast ice around Baffin Bay in fall, winter and spring, especially on the Greenlandic (east) ends of their passages A, B, and C. All of these points above make the comparison of these presented data in Baffin Bay to the data presented from Fram Strait a bit of a stretch to this reviewer. I appreciate that Kwok 2007 makes a comparison of the annual volume export he calculated for Baffin Bay and Davis Strait to the annual export of sea ice through Fram Strait, but it's my opinion that Kwok made that comparison in his 2007 work to simply highlight the amounts of sea ice exported southward in the two regions, and not to compare the processes or sea ice stages of development that typically are exported in the two areas

which are not physically similar.

The significance of this work is Poor. I think especially a re-worked version of this paper could improve our understanding of sea ice flux form this important region, but as it stands this work falls short of improved scientific understanding of the region and its relevant physical oceanographic, atmospheric, and/or sea ice processes. The analyses are comparative rather than investigative, and their presentation is in the style of a data report.

The presentation quality of this work is Fair. The figures are too numerous, and each conveys too little information. The authors convey too much data in Tables, while the text does not explain either the figures or tables more than superficially. This work is way too long, and clarity of direction is missing from the objectives onward through the results and discussion sections. The reason for the decadal period break at 2000/2001 is not obvious to this reviewer. It seems a break of convenience rather than scientific reasoning. Little in the way of conclusions is presented in that section of the text, though "...A comparison shows that our SIF estimates are reasonable consistent with previous results" (page 25 Line 8) is encouraging, it's not a conclusion. There are two more conclusions stated in Lines 9-11 of Page 25, but their validity is brought into question for this reviewer by the changing definition in the work (Oct or Nov to May depending on the section I read). The conclusions based on the three defined Gate locations make this reader interested in why their locations were chosen (especially given my previous note on the position of the northernmost Gate)? The last two paragraphs of the conclusions section are statements, which cannot be concluded as a result of the new work presented here.

Specific Comments: P1. L12: This sentence should make some reference to sea ice melt? L14: Why three passages, this really isn't useful information in the abstract given there is no geographical reference to their actual positions. L20: Causation is not shown in the work. Could the decline in SIM be a result of the fact that the SIM data are calculated in part from the SIC data? L24: Unclear what you mean here.
L28FF: The data exist to determine if the sea ice is in free drift in Baffin Bay and Fram Strait, but quantification is not attempted. Not sure why the authors insist on acronyms, especially for Fram Strait?

P2. L7: What is the potential impact? L9: Does outflow imply melt?

P3: L1: "passages" is a poor word choice, my opinion only. L2: Is there an effect to the trends/forcing? L5: What is the point of the comparison to Fram Strait? L7: "preferred..." Whose preference? Why? L28: I wonder what causes the discrepancy in the SMMR and SSM/I records with respect to sea ice motion data?

P4. L5-9: this whole section is worded like the authors actually did this processing? L10: Maybe use the whole words? Especially in a Section Heading? L16: Citation required? Figure 1: The position of Gate A is too far south to allow for quantification of ice flux from Lancaster Sound. I'm not really sure of the point of Gates B and C unless there's some quantification of melt? There's no scale for the magnitude of the vectors displayed on the figure.

P5. L8-10: What is the mechanism for ice motion through Baffin Bay with the NAO atmospheric patterns? Especially considering the height of the Greenland Ice Sheet that separates the Icelandic low from Baffin Bay? Does it even make sense that the NAO should drive ice drift in Baffin Bay? Some justification of this use of the NAO should be made? L17-24 and Table 1: Why order the Gates A, C, B, in the explanation of their positions? Why are all their lengths different in the text and the table? Seems like those lengths should be consistent?

P6. L2-3: Is this assumption valid? L11: Maybe call the Gates "North", "Mid", and "South" if you're going to continue using three so their positions are immediately apparent to the reader? L14, L16: Here's where I first noticed that the months used for "winter" change constantly through the work. Nov-May vs. Dec-May? There's got to be a consistent set of winter months used I think? This really reduced my trust in the analysis presented. L18: Where is the Cuny Gate in relation to Gates B, or C?

P7. Figure 2: The lines, colours are too hard to discern. Now winter is Nov-May. L12: Now winter is Oct-May? There are a bunch of typos in this paragraph. L13: It'd be nice to see the sea ice concentration data? L14: Where is the wind forcing data?

P8. Figure 3: The previous figure was in cm/s, now we've changed to km/day, and we have a legend for the magnitude of the quivers. Maybe add the Gates to the figure? It would be nice to see the actual sea ice concentration? Delineate the fast ice? What portion of the sea ice motion in Baffin Bay is driven by ocean currents? L3: Doesn't the data exist to determine this? Even the magnitude of the gradient? L6: Could the pattern not be visible due to your use of monthly averages? Seems like the higher the ice concentration the more likely that ice motion events might be temporally discrete due to the magnitude of the forcing required?

P9. L5: Why 4c before 4b? Figure 4: I don't understand why the comparison to Fram Strait. Probably should specify the Gates in the figure caption? L16-17: Why not quantify this? Also now winter is Dec-May?

P10. L7: There seems no scientific reason to break at 2000? L9: Now December is in the autumn? L15: What use the sci.notation? Why not just write -21km2/day? L31: Not really a decadal change, it's a change between two decades.

P11.L12: Now winter is Nov-May

P12. L10-11: This is not a result of this work and is left unsubstantiated.

P13. L4: Does this mean that the trend is not significant? L6: Shouldn't the "loose-ness" of the ice pack, that is it's internal stress, be quantifiable with the data here? L8: I don't know where this statement comes from. It' unsubstantiated by the work presented.

P14. Table 1: There's a better way to graphically display the pertinent/important parts of this Table. You're asking the reader to do too much work to understand your analysis.

P15-16. Figures 8, 9, 10: These figures are to small, to hard to read. Why use the

three different significance levels? Figure 10: the y-axis keeps changing, makes it hard to compare within the columns.

P17. L3: The increasing sea ice motion trend is not caused by a positive sea ice flux trend; you've put the cart before the horse here. L12: now winter is from Oct-May

P18. L3: What's the point of this section? L4: What does this sentence actually mean? Maybe this is some reference to melt within an area between two Gates?

P20. L15FF: What about the source of the exported sea ice? Smith Sound? Grown in the Northwater? From the CAA through Lancaster Sound? Surface winds in Baffin Bay are tricky to model because of the elevation of the Greenland ice sheet and the CAA islands? How well do these model winds actually represent reality? L16: What do all these acronyms mean? L20: Surface winds towards the southeast are northwesterly winds.

P21. L1: Winter is Oct-May now.

P22. This whole page seems like conjecture, it should all actually be borne out by some analysis. Kind of seems like the authors are listing possibilities rather than elucidating processes.

P23. L16-17: isn't the NAO calculated from the pressure difference between these two atmospheric phenomena? Hurrell 1995?

P24. L16-18: The internal stress of the sea ice pack should be able to be approximated at least. This sentence seems like conjecture as is.

P25. Figure 17. Any actual information in these panels is undiscernable due to their size.

There's no mention of the North Water Polynya in the conclusions section. L15-20: I think the statements in this paragraph remain unsubstantiated by this work. What about fast ice extent? The last two sentences are not conclusions of this work. I don't really

understand why you've listed surface winds and sea level pressure because they're two sides of the same coin, same goes for SAT and SST?

P26: These are not new results. As indicated in part by your reference to Kwok's papers. The last sentence of this paragraph is not a conclusion that is supported by the work presented.

---

## Referee Comment (RC3) · Anonymous Referee #3 · 31 Aug 2018

Summary

The paper investigates sea ice area flux through Baffin Bay from 1988-2015, using satellite observations of sea ice motion and sea ice concentration (NSIDC). In particular, the authors calculate the ice area flux at three gates in the north, middle and south of Baffin Bay. They evaluate variability and trends of ice area flux as well as links to sea level pressure (SLP) and the North Atlantic Oscillation (NAO) index.

General Comments:

I don't really see that the paper is following a particular thread. The formulation of the goals in the introduction is very brief and general. The authors should make clear if this is rather a method paper, introducing a new data set, or a scientific study to investigate

sea ice fluxes and related processes in the Baffin Bay.

I find many motivations and conclusions questionable. For example, what is the motivation to compare the derived Baffin bay fluxes with the fluxes in the Fram Strait? I don't see why this is relevant. At most, one could compare the net fluxes of different gates in the Arctic and estimate the total sea ice export. But the mechanisms in both regions are very different. The Fram Strait ice flux is characterised by multiyear sea ice, which is advected by the transpolar drift and exported through the Fram Strait. In contrast, the Baffin Bay ice fluxes are characterised by first-year ice in winter, when no multiyear ice is exported through the Nares Strait due to an ice bridge. During summer, this ice bridge is collapsing, and multiyear ice can be exported through the Nares Strait. These processes are not well explained in the paper, but are very relevant to understand ice fluxes in this region.

Another debatable point is the decadal change around the year 2000. This seems arbitrary. The interannual variability seems to be quite substantial and therefore I don't think that there is a significant change in SIF between those particular decades (see Figure 2).

Another major concern is the presentation. There are too many figures, sometimes of low quality, and with too little significance. I strongly recommend to revise the figures in order to better support the findings and main messages of the paper. For example, Figures 8, 9 and 10: It is neither explained the meaning of the lower case letters in the brackets, nor the meaning of the rows (i guess the different gates + Fram Strait?).

I would also suggest to better separate results and discussions. In the results section, findings are often discussed.

Considering all these concerns, I suggest very substantial revisions. Actually, I think that many parts of the paper need to be rewritten, and also the analysis and conclusions need to be reconsidered, before it may be suitable for publication.

[Figure]

Detailed Comments:

P3L1: What is the motivation to consider three passages at the chosen locations? Please add some explanation.

Figure 1: There are Chinese (?) letters in the figure.

P5L21: What do you mean here (and in other places) with "grid"? Do you mean grid cells, pixels? This needs to be explained better, i.e. use grid cells or pixels.

P10L6-7: can you proof that this change is significant? In view of Figure 2, I would doubt. See my major concern above.

Figure 2: Make the figure larger, please!

Figure 3: The scaling of the arrows changes between 0.1 km/day and 10 km/day. Please use a uniform scaling. Otherwise, the different months are hard to compare.

Figure 7: What are a, b, c and d? There is no information in the figure caption.

Figure 14: Make the figure larger, please!

P17L3: "For passages A and B, the increasing SIM trend (Figure 10b and f) is primarily caused by a positive SIF trend" ... This doesn't make sense. SIF is derived using SIM.

P24L4-5: "However, Figure 16 suggests that the monthly SIF is only slightly 5 correlated with the NAO index for the three passages through Baffin Bay (R = 0.23~0.32)" ... Why should they be correlated? See my major concern above.

---

## Author Comment (AC1) · 30 Nov 2018

Response to reviewer 1

This paper tries to monitor the sea ice flux (SIC) changes in Baffin Bay from 1988 to 2015 using the NSIDC sea ice concentration (SIC) and sea ice motion (SIM) datasets. The authors also try and link these changes in SIF to climate variables from the NCEP-NCARR dataset. Thought the datasets used for this study are very relevant and results can be very relevant to understanding the climate variability of sea ice conditions in Baffin Bay, after reading the paper, I do not think the authors understand the sea ice conditions in Baffin Bay and the drivers of the sea ice fluxes properly.

Response: Following the suggestion, a introduction of the sea ice conditions in Baffin

[Figure]

Bay is depicted in Section 3 in the revised manuscript. Section 3.1 gives a brief description of sea ice coverage. Section 3.2 make a introduction about the general sea ice drift pattern. Section 3.3 presents the trends in concentration and sea ice motion in Baffin Bay. These facts complement the understanding of the changes and the driving forcing of sea ice flux appropriately for different gates as seen in the discussion (Section 5).

My first concern is why the authors keep comparing the SIF of Baffin Bay with the SIF of Fram Strait. The ice regimes of those two regions are very different. The source of the SIF from Fram Strait comes directly from the Arctic Ocean and a large portion of the ice that flows there is comprised of multi-year ice all year round. In the case of Baffin Bay, the ice flows from Lancaster Sound and Nares strait in the summer months and is a mix of multi-year ice and first-year ice but in the winter, the ice mainly comes from first-year ice generated in polynyas of Lancaster Sound and the northern part of Baffin Bay. A major driver in Baffin Bay is the North Water Polynya (NOW) and nowhere do the authors mention this. Also, that polynya exists because of the ice bridge that is created in Nares Strait in the winter which blocks the inflow of thicker multi-year ice from the Arctic Ocean. Fram Strait and Baffin Bay are thus not comparable in terms of SIF.

Response: Thanks for the insightful suggestions. Recognizing that Fram Strait is a totally different areas for sea ice export, in the revised manuscript we remove the comparison with the flux in Baffin Bay. Considering the fact that North Water Polynya (NWP) is the primary ice source to the Bay during cold-freezing periods, we altered the positions of the North fluxgate (Figure 4) to further higher latitudes and quantify the sea ice inflow components originating from Lanscater Sound and NWP. The mitigating effects of sea ice flux though Nares Strait owing to the appearance of ice bridge (arch) during winter period are demonstrated in CIS maps (Figure S1) and reiterated in Section 2.1.3. Quantitatively, our ultimate estimates showing that 78%∼85% (Section 5.1.3) of the ice inflow through the North Gate consists of ice grown in NWP. This fact outlines

the recurring polynyas during cold seasons as a major contributor to ice area entering into the northern Baffin Bay.

Another issue is the separation between before and after the year 2000. The selection of the year 2000 seems arbitrary to me. Many climate change studies (including the IPCC report) state that there was a climate shift around 1998. Why did the authors not look at before and after 1998? Why don't there results show this climate regime shift (it should appear in the NCEP-NCARR dataset).

Response: In the revised manuscript, we focus on the issue of interannual variability and the long-term trends of sea ice area flux over the past nearly four decades and discuss the primary causes for the observed interannual variations and trends. As suggested by the reviewer, the inter-period changes, as previously separated by 2000, is not proper and excluded in the revision. Instead, we concentrate on the variability of the month-to-month ice transport of the past four decades (Figure 10) as well as the long-term trends (Figure 11). In particular, the climate changes associated with a warmer Northern Hemisphere over time, as shown in the NCEP-NCAR SAT data (Figure 16), are related to a thinner ice cover (Figure 15) and thus the increased area flux (Figure11). Also, some results can be explained by the sea ice conditions themselves. For example, the low SIM values on the Greenland coast in the winter months can be explained by the ice conditions. The sea ice along the coast is land fast, i.e. attached to the Greenland coast and does not move (note: the extent varies each year). I would strongly recommend that the authors visit the Canadian Ice Service website (https://www.canada.ca/en/environment-climate-change/services/iceforecasts-observations/latest-conditions.html) and specifically their 30-year ice atlas (https://www.canada.ca/en/environment-climate-change/services/ice-forecastsobservations/latest-conditions/climatology/30-year-climatic-atlases.html) to better interpret the SIF results.

Response: This piece of comment is vital to this study. Accordingly, we inspect the CIS maps for the ice conditions in Baffin Bay. Especially important is the knowledge of

spatial distribution of land-fast ice around the coasts of the bay, as provided in the CIS ice atlas. The land-fast ice extent benefits our study in that it helps to identify the zero flux grid cells which generally appears around the endpoints of one fluxgate adjacent to coasts. Secondly, it provides a thread to understand the slow ice motion, as introduced by reviewer, in the region attached to the Greenland coast (Figure 2, red arrows).

Overall, I would not reject this paper since it is very relevant to the studied field of climate change but I suggest major reviews after the authors better describe the region of interest and its different drivers. I would suggest to remove the comparison with Fram Strait as they don't have the same ice regimes. I would also study a bit more in detail the ice conditions which can be obtained on a weekly basis from the Canadian Ice Service in order to improve the interpretation of the results.

Response: As suggested by the reviewer, a major review is conducted by the authors. We remove the comparison with ice export through Fram Strait for its totally geophysical setting for sea ice. More information about the ice conditions in Baffin Bay is given (Section 3), such as the appearance and distribution of ice bridges in Nares Strait (Figure s1), the annual sea ice extent of the bay and coverage of NWP (Figure 6), the typical current circulation systems (Figure 1), and so on.

Specific comments: In the figure caption, I suggest adding more detailed descriptions of the figures. Specify what the a), b), c) and d) subfigures are etc. For Figure 7, what is the reference to generate the anomaly map? Usually it's a specific period that is used but it was not specified in the text. Also, what are the units if any for this anomaly map?

Response: Revised as suggested. To make it clear and direct to readers, we add captions to Figures with multiple subfigures (such as Figure 10). Figure 7 in the original submission has been removed in the new version of manuscript. Moreover, in the revision, we specify for each examined period or time interval anywhere mentioned in case of any ambiguity. For instance, we analyze the sea ice area flux for the

four different seasons including Winter (December-February), Spring (March-May), Summer (June-August), and Autumn (September-November). In some places , we refer the cold-period to include winter and spring or warm-period to cover summer and autumn seasons, if no particular annotation is given (See 5.3.2). Note that in the comparison of accumulated sea ice flux between our and previous studies (Figure 5), the cold period spans a time interval from Nov to May. Therefore, we try every effort to give an additional explanation about the time range to ensure that the alternatively used time period will not confuse the readers.

Please also note the supplement to this comment:
https://www.the-cryosphere-discuss.net/tc-2018-136/tc-2018-136-AC1-supplement.pdf

---

## Author Comment (AC2) · 30 Nov 2018

In reference to the review criteria provided by The Cryosphere (https://www.thecryosphere.net/peer_review/review_criteria.html), I submit the following general commentary on this work. The scientific quality of this work is Fair. The purpose of the work is articulated, however the objectives laid out in the second-to-last paragraph of the introduction section do not strike this reviewer as testable hypotheses. They are rather statements resulting in the paper becoming something of a data report. The analyses presented are somewhat perfunctory, and potentially these points indicate a lack of understanding of the physical atmospheric and/or oceanic processes acting on the surface (ocean or sea ice) within Baffin Bay.

[Figure]

Response: Thanks for the overall insightful comments. In the revised manuscript, we make it clear that our primary attempt is to quantify the sea ice inflow and outflow components through the key fluxgates of Baffin Bay and to investigate the related causes for their variability and trends. According the suggestions of reviewers, the fluxgates have been reslected to account for different sources of ice (Figures 1 and 4). In addition to the interannual variability, the trend of sea ice area flux for different gates are discussed and possible causes are analyzed. Compared with the first submission, we try to present our study in a form of investigation rather than a data report. We present more knowledge with regard to the atmospheric and/or oceanic process. For instance, the general circulation pattern of ocean currents in the bay is shown in Figure 1. The description of NWP (Figure 6) and associated ice bridges in Nares Strait are given (Figure s1). Additionally, the fast-ice extent is presented in Figure 2 to discriminate the grid cells with a zero flux and interpret the slow ice motion in the areas nearby the coast. For the first time, the inflow and outflow components through Baffin Bay over the past nearly four decades (1978/79-2016/2017) are quantified (Sec.4, Figures 10 and 11). Further investigation suggests that, the warmer atmosphere are found to be a main driver to the increased sea ice motion as well as the area flux to a thinner ice floe in the bay (Figures 15 and 16). Therefore, in the new submission we made efforts to improve the understanding of the variability and trends of Baffin Bay sea ice area flux associated with inflow and outflow components (See the discussion section, Sec.5) for more details).

The first mention of the hemispherically significant North Water Polynya does not occur until Line 6 on page 23 of this work. In this vein, it's not really explained that the sea ice in Baffin Bay may be as a result of import from the Lincoln Sea as a result of motion through Nares Strait, Kane Basin, and then Smith Sound before it might either enter Baffin Bay, or encounter an ice-bridge blocking inflow to Baffin Bay from the north. The annual and seasonal presence/absence of this very important ice-bridge feature and the amount of MYI imported from the North is not explicitly investigated in this work.

Response: As North Water Polynya (NWP) is an important scenario in the northern Baffin Bay (Figures 1 and 6). In this study, we outline its importance and location not only in the introduction section, but also in the description of sea ice condition in Section 3.1 (Figure 6, panel for April). Moreover, the contribution to the sea ice inflow into Baffin Bay is quantified. The presence and absence of ice-bridge in Nares Strait are illustrated in Figure S1 and its importance to sea ice flux stoppage is discussed in the relevant texts. Anyway, the significance of NWP and ice bridge are reiterated in the new submission. The importance and effects of sea ice bridge to block ice inflow are outlined in the revised manuscript (Section 2.1.3). Figure S1 gives a typical case of ice bridge formed in Nares Strait. However, due to the coarseness of NSIDC ice motion data (25km), compared to smith sound (∼30 km), it is unlikely to accurately estimate sea ice flux in such a narrow gate. For narrow fluxgate like Smith Sound, land contamination would be a server problem to satellite observations. Instead, with reference to a published result of between 1996/97 and 2008/2009 about ice flux via Nares Strait that was derived high-resolution satellite observations (such as SAR in Kwok (2007), with a spatial resolution of several hundred meters), we get an estimate of ice grown in NWP. That is, by subtracting the inflow from Nares Strait and inflow from Jones Sound from the ice inflow across the North Gate, the part of sea ice grown in NWP is then obtained as, on average, 78%∼85% of the inflow through the North Gate

First-year stages of development of sea ice in Baffin Bay may then also have been grown in place within the North Water and exported southward, depending on the formation of the ice-bridge, and the amount of ice in the Bay, which has been imported from the Canadian Arctic Archipelago (CAA), mostly through Lancaster Sound.

Response: In the revised manuscript, the sea ice area inflow to the Baffin Bay and ice produced from NWP are both quantified. Please see Section 5.1 for the investigation of the possible ice sources for sea ice entering into the Baffin Bay.

Since the authors chose a position for their northernmost 'Gate A' south of Lancaster Sound, the flux of sea ice from the CAA in the west, or from Smith Sound in the north,

cannot be differentially discerned. Other important considerations that occur to this reviewer include sea ice melt during transport southward within Baffin Bay, especially considering that new and young stages of development (<30cm thick) may be grown and exported from the North Water Polynya.

Response: Following the suggestions and in order to discern the diverse Baffin Bay sea ice inflow ice sources, we redefined the North Gates to a further north location (Figure 4). Examination of sea ice inflow through the North Gate reveals that NPW is the main contributor of inflow (Section 5.3) and a smaller fraction of inflow come from Nares Strait (Section 5.1) and/or Jones Sound. In addition, sea ice inflow through the Lanscater Sound is also obtained (Section 5.2). Meanwhile, taking into account the outflow of sea ice via the South Gate, we find a net gain of sea ice between the regime of ∼65°N and ∼75°N within the bay during cold periods (winter and spring) and a net loss during warm melting period (autumn and summer). The addition of sea ice is mainly through the freezing mechanism whereas the ice loss is caused by enhanced melt during transport southward within the Bay. Please refer to section 4.2 for more information.

Finally, it might be that the authors have not accounted for the presence of fast ice around Baffin Bay in fall, winter and spring, especially on the Greenlandic (east) ends of their passages A, B, and C. All of these points above make the comparison of these presented data in Baffin Bay to the data presented from Fram Strait a bit of a stretch to this reviewer. I appreciate that Kwok 2007 makes a comparison of the annual volume export he calculated for Baffin Bay and Davis Strait to the annual export of sea ice through Fram Strait, but it's my opinion that Kwok made that comparison in his 2007 work to simply highlight the amounts of sea ice exported southward in the two regions, and not to compare the processes or sea ice stages of development that typically are exported in the two areas which are not physically similar.

Response: In the new submission, land-fast ice distribution is discerned from Canadian Ice Service (CIS) atlas (as shown in Figure 4). According to the suggestions, we

remove the comparison with sea ice area flux through Fram Strait. The fluxgates are reselected. The North Gate and Lanscater Sound are chosen to stand for ice inflow while the South Gate is selected to represent the ice outflow via Baffin Bay. The middle gate is not useful to convey new knowledge and not kept. The North Gate is different from the first submission for its relocated place toward further north. This gate is designed to provide valuable information about ice inflow from different ice sources, including Jones Sound, Nares Strait, as well as NPW.

The significance of this work is Poor. I think especially a re-worked version of this paper could improve our understanding of sea ice flux form this important region, but as it stands this work falls short of improved scientific understanding of the region and its relevant physical oceanographic, atmospheric, and/or sea ice processes. The analyses are comparative rather than investigative, and their presentation is in the style of a data report.

Response: Compared with the first submission, we try to present our study in a form of investigation rather than a data report. We present more knowledge with regard to the atmospheric and/or oceanic process. For instance, the general circulation pattern of ocean currents in the bay is shown in Figure 1. The description of NOW (Figure 6) and associated ice bridges in Nares Strait are given (Figure s1). The fast-ice extent is presented in Figure 2 to discriminate the grid cells with a zero flux and interpret the slow ice motion in the areas nearby the coast. For the first time, the inflow and outflow components through Baffin Bay are quantified for the past nearly four decades (1978/79-2016/2017) (Sec.4, Figures 10 and 11). Further investigation suggests that, the warmer atmosphere are found to be a main driver to the increased sea ice motion as well as the area flux to a thinner ice floe in the bay (Figures 15 and 16). Anyway, in the new submission we made every effort to advance the understanding of the variability and trends of Baffin Bay sea ice area flux associated with inflow and outflow components (See the discussion section, Sec.5) for more details).

The presentation quality of this work is Fair. The figures are too numerous, and each

conveys too little information. The authors convey too much data in Tables, while the text does not explain either the figures or tables more than superficially. This work is way too long, and clarity of direction is missing from the objectives onward through the results and discussion sections. The reason for the decadal period break at 2000/2001 is not obvious to this reviewer. It seems a break of convenience rather than scientific reasoning. Little in the way of conclusions is presented in that section of the text, though ": : :A comparison shows that our SIF estimates are reasonable consistent with previous results" (page 25 Line 8) is encouraging, it's not a conclusion. There are two more conclusions stated in Lines 9-11 of Page 25, but their validity is brought into question for this reviewer by the changing definition in the work (Oct or Nov to May depending on the section I read). The conclusions based on the three defined Gate locations make this reader interested in why their locations were chosen (especially given my previous note on the position of the northernmost Gate)? The last two paragraphs of the conclusions section are statements, which cannot be concluded as a result of the new work presented here.

Response: Based on the suggestions, we remove the numerous figures and Tables that convey little usefule information. The main objective of this study is clarified in the Introduction section, including the quantification of sea ice inflow and outflow of Baffin Bay, the examination of the variability and trends of ice area flux, as well as the investigation of causes for the observed trends in ice motion and area flux. We remove the discussion of area flux associated with climate change with a break of decadal period at 2000/2001 since this simplified partition has no clear geophysical implications rather than for the convenience of calculation. Rather, the month-to-month variability of sea ice area flux across the fluxgate for different decadal periods are given in Figure 10 (Sec. 4.1). Throughout the revised manuscript, the definition of seasonal and annual fields, as in reference to sea ice area flux, are clarified in the associated texts, with Winter spans from December to February, spring (March-May), summer (June-August), autumn (September-November), and the annual cycle (September-next August). We remove the middle gate since the little useful information is conveyed by

the comparison with other gates. Besides, comparing with the first submission, the location of the fluxgates to study the sea ice area flux are relocated, especially for the northward shift for the North Gate (Figure 4). The redefinition of the North Gate is favorable to the quantification of ice production in NWP (see 5.1.3 for more details), the well-known recurring polynyas in the northern Baffin Bay.

Specific Comments: P1. L12: This sentence should make some reference to sea ice melt? L14: Why three passages, this really isn't useful information in the abstract given there is no geographical reference to their actual positions. L20: Causation is not shown in the work. Could the decline in SIM be a result of the fact that the SIM data are calculated in part from the SIC data? L24: Unclear what you mean here.

Response: P1. L12: we have rewritten the sentence to refer the exported sea ice from Baffin Bay as one of important solid fresh waters input to the seas downstream. L14: The passages are redefined in the new submission and renamed with a reference to its location: North Gate, South Gate, and Lanscater Sound, etc. L20: The causation of the increased SIM is presented in the revision. L24: The ambiguous sentence has been removed.

L28FF: The data exist to determine if the sea ice is in free drift in Baffin Bay and Fram Strait, but quantification is not attempted. Not sure why the authors insist on acronyms, especially for Fram Strait?

Response: This sentence is removed since the comparison with Fram Strait conveys limited knowledge. The acronyms 'FS' for Fram Strait is not saved in the revision.

P2. L7: What is the potential impact? L9: Does outflow imply melt?

Response: Since outflow does not necessarily represent melting, we remove the sentences.

P3: L1: "passages" is a poor word choice, my opinion only. L2: Is there an effect to the trends/forcing? L5: What is the point of the comparison to Fram Strait? L7: "preferred:

: :" Whose preference? Why? L28: I wonder what causes the discrepancy in the SMMR and SSM/I records with respect to sea ice motion data?

Response: L1: we use fluxgate or gate instead of passage. L2: There is not an effect to the trend from the atmosphere (Section 5.1 or Figure 14) and the causes for the trend are examined with respect to ice thickness changes (Section 5.3). L5: The comparison is removed. L7: The analysis with large-scale atmospheric index is not held in the revision according to suggestions of the reviewer. In the revision, connections with regional atmosphere variability, with reference to cross-gate SLP difference, is taken as an important predictor for the variability in Baffin Bay sea ice area flux. L28: The sea ice concentration is available every other day for the period of SMMR which would bring a discrepancy. A temporal interpolation method is used to fill the gap and daily SIC is obtained for the period Nov 1978 to July 1987. Thereby we can extend the whole study time series to 1978/1979-2016/2017.

P4. L5-9: this whole section is worded like the authors actually did this processing? Response: The relevant reference has been added in the revision.

L10: Maybe use the whole words? Especially in a Section Heading? Figure 1: The position of Gate A is too far south to allow for quantification of ice flux from Lancaster Sound. I'm not really sure of the point of Gates B and C unless there's some quantification of melt? There's no scale for the magnitude of the vectors displayed on the figure. Response: L10: the whole words is used in all Section Heading. The position of the North Gate is moved to further north and the ice flux from Lanscater Sound is quantified. Gates B is removed as no information of melt can be derived. The scale of the magnitude of ice motion vectors are added on the relevant figures. The fluxgates are rearranged in the revision (Figure 4). The North Gate and Lanscater Sound are chosen to assess ice inflow while the South Gate is selected to study the ice outflow via Baffin Bay. The middle gate is not useful to convey new knowledge and not kept in the revision. The North Gate is different from the first submission for its northwardly relocated place. This gate is designed to provide valuable information about ice inflow from different ice sources, including Jones Sound, Nares Strait, as well as NPW (Section 5.1.3).

P5. L8-10: What is the mechanism for ice motion through Baffin Bay with the NAO atmospheric patterns? Especially considering the height of the Greenland Ice Sheet that separates the Icelandic low from Baffin Bay? Does it even make sense that the NAO should drive ice drift in Baffin Bay? Some justification of this use of the NAO should be made?

Response:L8-10: Based on the suggestions, we remove the analysis with regard to the connections between NAO and sea ice drift in Baffin Bay.

L17-24 and Table 1: Why order the Gates A, C, B, in the explanation of their positions? Why are all their lengths different in the text and the table? Seems like those lengths should be consistent?

Response: The gates have been redefined and relocated for A (the North Gate), and the middle gate (B) is removed in the revision. Gate C is renamed as the South Gate. The reasoning to rearrange the fluxgate is mentioned above. In addition, we examine the texts and table to ensure a consistent use of the length for the Gates.

P6. L2-3: Is this assumption valid? L11: Maybe call the Gates "North", "Mid", and "South" if you're going to continue using three so their positions are immediately apparent to the reader?

Response: L2-3:The assumption is widely adopted in previous studies and especially one may refer to Kwok's studies associated with sea ice area flux. L11: The Gates are renamed following the suggestions to specify the geographic information.

L14, L16: Here's where I first noticed that the months used for "winter" change constantly through the work. Nov-May vs. Dec-May? There's got to be a consistent set of winter months used I think? This really reduced my trust in the analysis presented. L18: Where is the Cuny Gate in relation to Gates B, or C?

Response: L14, L16: In the revision, four seasons are considered, including the winter (Dec-Feb), Spring (Mar-May), summer (Jun-Aug), and autumn (Sep-Nov). Also, the cold period is referred to include the winter and spring months while the warm period spans the summer and autumn months, if no particular annotation is given. These definitions hold constantly in the new submission unless particular annotation is shown. L18: Cuny's estimate is related to the South Gate in the revision (Figure 4). In Figure 5, the comparisons are confined to a time span from Nov to May in the following year.

P7. Figure 2: The lines, colours are too hard to discern. Now winter is Nov-May. L12: Now winter is Oct-May? There are a bunch of typos in this paragraph. L13: It'd be nice to see the sea ice concentration data? L14: Where is the wind forcing data?

Response: The figure is modified to make a discernable color. To neatly convey information about sea ice drift pattern in Baffin Bay, this part has been reworked in the new submission (see section 3.2)

P8. Figure 3: The previous figure was in cm/s, now we've changed to km/day, and we have a legend for the magnitude of the quivers. Maybe add the Gates to the figure? It would be nice to see the actual sea ice concentration? What portion of the sea ice motion in Baffin Bay is driven by ocean currents? L3: Doesn't the data exist to determine this? Even the magnitude of the gradient? L6: Could the pattern not be visible due to your use of monthly averages? Seems like the higher the ice concentration the more likely that ice motion events might be temporally discrete due to the magnitude of the forcing required?

Response: Figure 3: The unit has been changed to km/day throughout the manuscript. The sea ice concentration changes of an annual cycle are shown in Figure 6. The portion of sea ice motion driven by ocean currents can be observed through the comparison with Figure 1 and Figure 6. Relevant explanations have been given in associated texts in the revision. Besides, the sea level pressure (SLP) fields are overlaid on Figure 6. The presence of SLP is helpful to distinguish the effects of wind forcing on facilitating

the sea ice motion ice motion in Baffin Bay.

P9. L5: Why 4c before 4b? Figure 4: I don't understand why the comparison to Fram Strait. Probably should specify the Gates in the figure caption? L16-17: Why not quantify this? Also now winter is Dec-May?

Response: Following the suggestions, the comparison with Fram Strait is removed and the figure captions as depicted in Figure 10 is specified. The season discrimination is redefined (see response above).

P10. L7: There seems no scientific reason to break at 2000? L9: Now December is in the autumn? L15: What use the sci.notation? Why not just write -21km2/day? L31: Not really a decadal change, it's a change between two decades.

Response: Based on the relevant suggestions, we removed this part of results.

P11.L12: Now winter is Nov-May P12. L10-11: This is not a result of this work and is left unsubstantiated.

Response: in the revision, the delayed ice freezing period due to warmer climate is refer to the published literature of Stroeve et al. (2014) (see P22. L21 in the revised manuscript)

L8: I don't know where this statement comes from. It' unsubstantiated by the work presented.

Response: the summer or autumn trend in ice flux is negligible and no further analysis is given in the revision.

P14. Table 1: There's a better way to graphically display the pertinent/important parts of this Table. You're asking the reader to do too much work to understand your analysis.

Response: The table is removed for a clear interpretation but rather a comprehensive figure is provided (Figure 14)

P15-16. Figures 8, 9, 10: These figures are to small, to hard to read. Why use the three different significance levels? Figure 10: the y-axis keeps changing, makes it hard to compare within the columns.

Response: Since these figures show little useful information, we remove them and integrate them into a comprehensive one (Figure 14). More details are given in the following associated texts.

P17. L3: The increasing sea ice motion trend is not caused by a positive sea ice flux trend; you've put the cart before the horse here. L12: now winter is from Oct-May

Response: Reformulated as suggested.

P18. L3: What's the point of this section? L4: What does this sentence actually mean? Maybe this is some reference to melt within an area between two Gates?

Response: This part is reworked and relevant explanation is reformulated in the revision, please see section 4.2 for more details.

P20. L15FF: What about the source of the exported sea ice? Smith Sound? Grown in the Northwater? From the CAA through Lancaster Sound? Surface winds in Baffin Bay are tricky to model because of the elevation of the Greenland ice sheet and the CAA islands? How well do these model winds actually represent reality? L16: What do all these acronyms mean? L20: Surface winds towards the southeast are northwesterly winds.

Response: The inflow components to Baffin Bay from diverse sources, Nares Strait, Lanscater Sound, or grown in the North Water Polynya are discussed in Section 5.1. Wind data is not used in the revision.

P22. This whole page seems like conjecture, it should all actually be borne out by some analysis. Kind of seems like the authors are listing possibilities rather than elucidating processes.

Response: In order to understand the associated air and physical processes, we investigate the connections of sea ice flux variability with cross-gate SLP difference (Sec. 5.2), and examine the linkage between ice motion and ice thickness through a preliminary simulation (Sec. 5.3).

P23. L16-17: isn't the NAO calculated from the pressure difference between these two atmospheric phenomena? Hurrell 1995?

Response: NAO is not considered in the revision based on the above comments of the reviewer.

P24. L16-18: The internal stress of the sea ice pack should be able to be approximated at least. This sentence seems like conjecture as is.

Response: The linkage to a faster movement of ice pack is mostly attributable to ice thickness decline and the analysis with internal stress is beyond our scope of this study.

P25. Figure 17. Any actual information in these panels is undiscernable due to their size. Response: This Figure is removed in the revision.

There's no mention of the North Water Polynya in the conclusions section. L15-20: I think the statements in this paragraph remain unsubstantiated by this work. What about fast ice extent? The last two sentences are not conclusions of this work. I don't really understand why you've listed surface winds and sea level pressure because they're two sides of the same coin, same goes for SAT and SST?

Response: The NWP is reiterated in the revision. For instance, one may refer to Figure 1 and 6 and associated texts. In particular, Section 5.1.3 also specify the contribution of sea ice grown in NWP. We kept well-demonstrated SLP and SAT fields, and remove the SST and winds. As suggested, SLP and winds (or SAT and SST), reflect the same climatic scenario.

P26: These are not new results. As indicated in part by your reference to Kwok's papers. The last sentence of this paragraph is not a conclusion that is supported by

the work presented.

Response: The conclusions are reworked and the associated unproven texts are eliminated.

Please also note the supplement to this comment:
https://www.the-cryosphere-discuss.net/tc-2018-136/tc-2018-136-AC2-supplement.pdf

[Figure]

**Supplement:**

[revised manuscript text omitted]
 final Envisat estimates are smoothed to a 25-km grid, and then spatially registered to the NSIDC data.

Two examples of Envisat ice motion fields, acquired on February 2007, are shown in Figure 2. One example covers a cyclonic circulation along the west coast of Greenland (red arrow) and the other is located in the area next to Davis Strait (blue arrow). Comparative results (Figure 3a) present a mean bias of -0.68 km/day in ice speed between the two records (i.e., slightly slower NSIDC) but a relatively large standard deviation of difference (3.11 km/day). Furthermore, there is a small average difference of 3.4 ° in vector angle (Figure 3b), indicating that the NSIDC motion is likely biased to the right. A large standard deviation exists in the difference of motion vector angle (38 °), which is mostly caused by data pairs for the slower Envisat motions of less than 3 km/day (Figure 3b). Despite these phenomena, the two estimates agree well as a whole, as indicated by the high correlation between them (R = 0.87).

[Figure]

**Figure 2.** Sea ice motion examples from NSIDC and Envisat. Gray arrows corresponds to the NSIDC data on 4~7 February 2007. The superimposed arrows, one for 4~7 February 2007 (red) and the other for 19~22 February 2007 (blue), denote the smoothed Envisat estimates. The in-phase fast ice extent is marked as purple polygons.

[Figure]

[revised manuscript text omitted]
 that are dependent on the thickness changes as we simulated. The remaining small part of ice motion acceleration can be explained by ice concentration decline. To summarize this section, the sea ice motion and area flux increases in Baffin Bay over the past four decades are mainly attributable to a thinner sea ice thickness which is primarily associated with the increase in surface air temperature. This is consistent with findings in the Arctic Ocean (Rampal et al., 2009; Spreen et al., 2011; Kwok et al., 2013).

**6. Conclusions**

. With satellite-derived sea ice parameters, we estimated the sea ice inflow and outflow through the key fluxgates of Baffin Bay. The record of sea ice area flux was extended to span a nearly 40-yr period from 1978/1979-2016/2017. On the basis of the estimates, the variability and trends of sea ice area flux through the three fluxgates (North Gate, Lancaster Sound, and South Gate) for different timescales (monthly, seasonal, and annual flux) are examined in detail. Large interannual variations are detected for the different flux fields. Moreover, significant increasing trends are identified for the annual ice flux for the three gates, with the primary contributions from those during winter (December-February) and spring (March-May).

The spatiotemporal differences are obvious for the sea ice flux through different gates. On average, there is an inflow though North Gate ($205.8 \times 10^3 \mathrm{km}^2$) and Lancaster Sound ($55.2 \times 10^3 \mathrm{km}^2$) and an outflow via South Gate ($394.3 \times 10^3 \mathrm{km}^2$). During cold seasons (winter and spring), the difference between inflow and outflow (i.e. inflow minus outflow) amounts to $-153.3 \times 10^3 \mathrm{km}^2$ and is largely replenished by new ice formed within the bay that is likely associated with the divergence

mechanism. For the warm period (summer and autumn), the sea ice inflows ($46.3 \times 10^3 \, \text{km}^2$) and outflows ($26.3 \times 10^3 \, \text{km}^2$) are small, pointing to a net ice area loss of $20.0 \times 10^3 \, \text{km}^2$ that is connected to melting process in the bay. This emphasizes that the Baffin bay serves as not only an area of ice source during cold periods and but also an area of ice sink during warm periods. The sea ice growth and melting processes could have vital influences on the ocean current properties in Baffin Bay. With regard to the diverse ice inflow sources into the northern Baffin Bay through the North Gate, the comparisons with published results seem to tally well with the fact that the majority of (about 75%~85%) of ice area originates from ice growth in NWP, in addition to the ice inputs via Nares Strait, Lancaster Sound, and Jones Sound.

The interannual variability of ice flux across the North and South Gates is in part linked to wind forcing associated with the cross-gate SLP differences, while the ice flow through Lancaster Sound is largely determined by orographic conditions. The trends for the three gates (North Gate: $38.9 \times 10^3 \, \text{km}^2/\text{de}$; South Gate: $82.2 \times 10^3 \, \text{km}^2/\text{de}$; Lancaster Sound: $7.5 \times 10^3 \, \text{km}^2/\text{
[revised manuscript text omitted]

Supplement figures:

[Figure]

Figure S1a. CIS map for sea ice category in April 2014. The young ice south of the two bridges are shown.

[Figure]

Figure S1b. CIS map of sea ice concentration in April 2014. The two ice bridges are shown. The corresponding RADARSAT imagery is displayed in the inset grey figure.

---

## Author Comment (AC3) · 30 Nov 2018

The paper investigates sea ice area flux through Baffin Bay from 1988-2015, using satellite observations of sea ice motion and sea ice concentration (NSIDC). In particular, the authors calculate the ice area flux at three gates in the north, middle and south of Baffin Bay. They evaluate variability and trends of ice area flux as well as links to sea level pressure (SLP) and the North Atlantic Oscillation (NAO) index.

Response: According to the comments of reviewers, we rework the study with the primary focus on the subject regarding the sea ice inflow and outflow through Baffin Bay (Figures 1 and 4). Hence, three fluxgates are considered in the revision, including the North Gate, South Gate, and Lanscater Sound. The Gates were renamed with

a reference to geographic location to facilitate the direct understanding of readers. Moreover, the inflow components from different ice sources is examined, including sea ice area flux from Smith sound (Nares Strait), Lanscater Sound, as well as ice grown in North Water Polynya (NWP) (Section 5.1). The interannual variability is connected to the cross-gate sea level pressure (SLP) difference (Section 5.2). The linkage with NAO is not held in the new submission, since reviewer 2# suggests a fragile relation between NAO and sea ice flow in Baffin Bay exists because the effects of NAO is likely kept away from Baffin Bay by the high elevation of Greenland ice sheets. Furthermore, causes for the increased sea ice area flux are related to the sea ice thickness decline associated with a warmer climate. Therefore, the new submission represents for a substantial revision of the initial work.

General Comments: I don't really see that the paper is following a particular thread. The formulation of the goals in the introduction is very brief and general. The authors should make clear if this is rather a method paper, introducing a new data set, or a scientific study to investigate sea ice fluxes and related processes in the Baffin Bay.

Response: In the new submission, we make a straightforward attempt of this study to investigate the sea ice inflow and outflow through Baffin Bay and related causes for the observed interannual variability and the trend in sea ice area flux over the nearly 40-yr time series record (Section 1).

I find many motivations and conclusions questionable. For example, what is the motivation to compare the derived Baffin bay fluxes with the fluxes in the Fram Strait? I don't see why this is relevant. At most, one could compare the net fluxes of different gates in the Arctic and estimate the total sea ice export. But the mechanisms in both regions are very different. The Fram Strait ice flux is characterised by multiyear sea ice, which is advected by the transpolar drift and exported through the Fram Strait. In contrast, the Baffin Bay ice fluxes are characterised by first-year ice in winter, when no multiyear ice is exported through the Nares Strait due to an ice bridge. During summer, this ice bridge is collapsing, and multiyear ice can be exported through the Nares

Strait. These processes are not well explained in the paper, but are very relevant to understand ice fluxes in this region. Another debatable point is the decadal change around the year 2000. This seems arbitrary. The interannual variability seems to be quite substantial and therefore I don't think that there is a significant change in SIF between those particular decades (see Figure 2).

Response: These comments are very insightful. Following the suggestions, the comparison with Fram Strait is not saved in the revision. The importance and effects of sea ice bridge to block ice inflow are reiterated in the manuscript (Section 2.1.3). For instance, Figure S1 gives a typical case of ice bridge formed in Nares Strait. However, due to the coarseness of NSIDC ice motion data (25km), compared to smith sound ($\sim$30 km), it is unlikely to accurately estimate sea ice flux in such a narrow gate. For narrow fluxgate like Smith Sound, land contamination would be a severe problem to satellite observations. Instead, in reference to a published result of between 1996/97 and 2008/2009 about ice flux via Nares Strait that was derived high-resolution satellite observations (such as SAR in Kwok (2007), with a spatial resolution of several hundred meters), we get an estimate of ice grown in North Water Polynya (NWP). That is, by subtracting the inflow from Nares Strait and inflow from Jones Sound from the ice inflow across the North Gate, the part of sea ice grown in NWP is then obtained. Based on the suggestions, the simply separation in 2000 do not have scientific implications. Rather, the results about the decadal changes with respect to the annual month-to-month variability for four different decades are described in section 4.1 and shown in Figure 10.

Another major concern is the presentation. There are too many figures, sometimes of low quality, and with too little significance. I strongly recommend to revise the figures in order to better support the findings and main messages of the paper. For example, Figures 8, 9 and 10: It is neither explained the meaning of the lower case letters in the brackets, nor the meaning of the rows (i guess the different gates + Fram Strait?). I would also suggest to better separate results and discussions. In the results section,

findings are often discussed. Considering all these concerns, I suggest very substantial revisions. Actually, I think that many parts of the paper need to be rewritten, and also the analysis and conclusions need to be reconsidered, before it may be suitable for publication.

Response: Thanks for these valuable suggestions. In the revision, we remove lots of figures and rewritten most of the parts. The results and discussion are further adjusted for a clearer separation. Indeed, the new submission represents for a reworked version of the study. The analysis and conclusions are considered after a prudent consideration.

Detailed Comments: P3L1: What is the motivation to consider three passages at the chosen locations?

Response: The fluxgates are reselected in the revision. The North Gate and Lanscater Sound are chosen to stand for ice inflow while the South Gate is selected to represent the ice outflow via Baffin Bay. The middle gate is not useful to convey new knowledge and not held then. The North Gate is different from the first submission for its relocated place toward a further north position. This gate is designed to provide valuable information about ice inflow from different ice sources, including Jones Sound, Nares Strait, as well as NPW.

Please add some explanation. Figure 1: There are Chinese (?) letters in the figure.

Response: Modified as suggested

P5L21: What do you mean here (and in other places) with "grid"? Do you mean grid cells, pixels? This needs to be explained better, i.e. use grid cells or pixels.

Response: Corrected as suggested

P10L6-7: can you proof that this change is significant? In view of Figure 2, I would doubt. See my major concern above.

Response: Please refer to the relevant response to the major concern

Figure 2: Make the figure larger, please!

Response: Modified as suggested. See Figure 5 in the revision

Figure 3: The scaling of the arrows changes between 0.1 km/day and 10 km/day.

Please use a uniform scaling. Otherwise, the different months are hard to compare.

Response: Modified as suggested. See Figure 7 in the new submission.

Figure 7: What are a, b, c and d? There is no information in the figure caption.

Response: This Figure is removed but other Figures with multiple panels are all added with the figure caption (For example, Figures 10, 13, and 14).

Figure 14: Make the figure larger, please!

Response: This Figure is removed

P17L3: "For passages A and B, the increasing SIM trend (Figure 10b and f) is primarily caused by a positive SIF trend" : : : This doesn't make sense. SIF is derived using SIM.

Response: This is a wrong sentence and has been reformulated.

P24L4-5: "However, Figure 16 suggests that the monthly SIF is only slightly 5 correlated with the NAO index for the three passages through Baffin Bay (R = 0.23âĹij0.32)" : : : Why should they be correlated? See my major concern above.

Response: The NAO effects are kept away due to the height of Greenland ice sheets. Therefore, we remove the concerning discussion of the linkage with it.

Please also note the supplement to this comment:
https://www.the-cryosphere-discuss.net/tc-2018-136/tc-2018-136-AC3-supplement.pdf

---

## Author Response (AR2)

Baffin Bay sea ice inflow and outflow:1978/1979-2016/2017

Reviewer 1

General comments:

The revised version of this manuscript is greatly improved. The comments from the previous reviews have been properly addressed. Only minor reviews remain.

Minor revisions:

P1. L.19-20: Change NWP acronym to the one commonly used for the North Water Polynya (NOW). NWP is used for North West Passage in the Arctic.

Response: The 'NWP' has been replaced by 'NOW' throughout the manuscript.

P.2 L.10: change "influences on" to "influence"

Response: Changed as suggested.

P.3 L.21: change "of necessary" to "necessary".

Response: changed as suggested

P.4 L.3-4: I would change this sentence to say that this study focuses on the period of November 1978 to February 2017. No need to say that this was the data available at the time of the study. Response: We reconstruct the sentence without emphasizing that the study period is in line with data time period available.

P.9 L.17: change "convoluted" to "convolved". This is the proper mathematics term.

Response: changed as suggested

P.17 L.5: Change "There sources" to "The sources that".

Response: changed as suggested

Reviewer 2

General Comments:

The paper has changed significantly and now focuses on sea ice area flux through Baffin Bay, using three gates (different ones compared to the first version). The authors also expand the time series, now including the full time span of the NSIDC ice motion data set starting in 1978. The comparison with the Fram Strait flux has been discarded.

Unfortunately, the revision does not include a track of the changes in the new version of the paper. But from what I found, it seems that major parts of the paper have been rewritten and restructured. This is acknowledged and I find the paper improved now, following a certain thread. Also the quality and compilation of figures has improved.

Response: Thanks a lot for the supporting comments above.

The major objective of the paper is to derive long term trends and variations in sea ice flux through Baffin Bay. Former studies (Cuny et al., 2005, Curry et al., 2014, Kwok, 2007) have presented sea ice flux retrievals in the Baffin Bay already, but only on shorter time scales. Still, my major objections are regarding the soundness of the study:

1. Evaluation of the sea ice motion: The authors write: "To assess the NSIDC data, a reference product of sea ice motion, which is visually retrieved from high-resolution (~100 m) Envisat

wide-swath (~450 km) observations, is employed". I wonder what is meant by "visually"? It should be based on a robust retrieval algorithm!? In any case, you should describe how you have retrieved the reference ice drift from the Envisat data in much more detail. Moreover, it is not well described how the comparison is performed.

Response: It's our mistake not showing the method that we adopted. Indeed, the SAR-derived sea ice motion product is obtained following the algorithm provided in Kowk et al. (1990) to track common feature on images in sequence. In the revised manuscript, we present the information about this. Additional knowledge to sort out the erroneous Envisat-derived sea ice drifts are also given. That is, ice motion with a speed of larger than 5% of the NCEP wind is discarded, since they are likely related to the tracking of weather features. Visually inspect is finally used to identify the possible remaining small portion of erroneous ice motion that are not excluded by the 'wind' rule. Then, if the sea ice speed or direction value of a grid lying out of mean  $\pm 2$ standard deviation of those of the surrounding eight grids in a  $3 \times 3$  matrix, it is treated as an invalid estimate. Inverse distance interpolation is used to give an estimate for grids without valid ice motion estimates. Please see the second paragraph in revised texts in P4. Moreover, more details about how to compare the two products are also shown. Please see the third paragraph in revised texts in P4 (i.e., between line 7 and line 16).

2. For the uncertainty of the NSIDC ice motion, you use the standard deviation of difference (3.11 km/day) between the NSIDC and Envisat-derived ice drift. I don't think that this accounts for the real uncertainty. There is a nice paper by Sumata et al. (2015), which provides an empirical error function for different sea ice drift products, including the NSIDC retrieval. It also shows that the uncertainty increases with higher drift speeds, i.e. underestimation of high drift speeds.

Response: Thanks very much for showing us study by Sumata et al (2015). In the revision, we adopted the empirical uncertainty function as given in Sumata et al. (2015) to estimate the NSIDC ice motion uncertainty, which is in related to sea ice concentration and drift speed variations. The uncertainty are recalculated and updated estimates are shown in Table 1. As suggested, this consideration helps to avoid the likely underestimation of uncertainty with respect to the high drift speed fields.

3. The study heavily relies on the NSIDC data set and assumes that this data set is consistent over the entire period from 1978-2018, despite the fact that many different sensors have been used for different periods to construct this time series. In order to calculate trends, the consistency should be validated and profound uncertainty estimates have to be provided.

Response: To verify the consistency of NSIDC ice motion for the periods before and after 1987, we compare the NSIDC data of the two periods to corresponding IABP buoy measurements. As shown in Figure 2, the bias and standard deviation of difference between NSIDC and IABP data are similar over the 1979-1987 (Figure 2a) and 1988-1994 (Figure 2b) periods. The agreement indicates that there is no significant difference for NSIDC ice motion data in the earlier 1979-1987 and latter periods.

4. The ice growth model used in section 2.3 is very basic and simple. For example, it does not handle variations in the snow cover?! Moreover, the derived decrease in thickness is then used to explain trends in the ice motion due to wind (see Eq 2). But the air-ice drag coefficient is not considered here neither, although it is an important parameter, resulting from the surface and bottom roughness of the sea ice. This approach seems to be very vague.

Response: We sincerely appreciate this insightful suggestion. IceBridge measurements point to that a declining trend exists in snow depth over first-year ice cover in Arctic Ocean, suggesting the potential impacts on ice thickness growth rate and thus long-term sea ice thickness changes. Therefore, using the

simple ice growth model to estimate sea ice thickness changes, by only taking into account the decreased freezing day degrees while neglecting the impacts owing to snow depth changes, may overestimate the declining trend of sea ice thickness. However, the relative consistency of increases in ice motion speed as derived from NSIDC data and our calculations using Eq.3 demonstrates that our estimates are credible (For more information, please see the last paragraph in Section 5.3.2). In the revised manuscript, we notice the readers to pay attention to this limitation when understanding our results.

Yes, variations of drag coefficient can influence the sea ice drift rate. A recent study indicates that air-ice and water-ice drag coefficients in Baffin Bay show increasing trends, by the order of approximately  $0.01 \times 10^{-3}$ /yr and  $0.06 \times 10^{-3}$ /yr, respectively (Tsamados et al., 2014). Considering this, we reassess the increase of sea ice motion due to the changes in air and water drag coefficients. The new estimates indicate that both sea ice thickness decrease and drag coefficients increases contribute a significant part to the sea ice motion increase in Baffin Bay area. (For more information, please see the last paragraph but one in Section 5.3.2).

Detailed comments:

P2L5: "a northward flowing West Greenland Current (WGC) along the Greenland coast 5 carries warm and salty water from the North Atlantic" ... Figure 1 says it is "cold"?

Response: It's a mistake in Figure 1and We removed it.

P4L5-16: How is the reference ice drift derived?

Response: We present the used algorithm as provided in Kowk (1990). Moreover, to sort out erroneous drift, we use NCEP reanalysis wind as a reference. That is, sea ice motion that are larger than 5% of wind are sored out, since they are likely related the tracking of weather features. See P4 Line 7-11.

Figure2: Are you comparing different periods? It looks like you compare Envisat ice drift from Feb 19-22 with NSIDC ice drift from Feb 4-7. You have to describe in much more detail how the comparison between NSIDC and Envisat ice drift is performed and how Envisat ice drift is derived.

Response: Figure 2 is modified. The derived sea ice drift in Feb 4-7 and 19-22 are separately presented, together with the NSIDC data in the corresponding periods. Please see Figure 3 in the revised manuscript.

Figure 4: Fast ice edges from the different periods are hard to distinguish. May be zoom in the different regions.

Response: Figure 4 is modified (see Figure 5 in the revised paper). Key regions with fast ice extent variations are enlarged to distinguish the fast ice extent variations during different months.

Section 2.2.2: The used gates in other studies seem to be at different locations. How does this affect the comparison?

Response: To investigate the estimates due to the difference locations of the gate, we derived the area flux through a gate further south as used in the previous studies near to the Davis Strait. Although the previously-used gate is narrower compared to the south gate used in this study, we found the faster ice drift compensate the difference in the sea ice area flux due to the flux width changes. Indeed, the area flux in the south gate (this study) and the Davis Strait gate (previous study) is in good agreement (not shown).

Figure 10: How do you explain that the mean sea ice area flux in the 1978-1987 period is so low compared to the later periods? In any case, this needs to be investigated in more detail (see my concern

about the consistency of the NSIDC data set).

Response: We compare the NSIDC data to IABP buoy measurements with data for the 1979-1987 and 1988-1994 periods. For more details, please see Figure 2 in the revised manuscript. Uncertainty scales (bias and standard deviation of the two ice speed difference) do not show remarkable difference over the earlier (1979-1987) and the later (1988-1994) periods.

Figure 12: unit - How do you derive "km" from the divergence of a velocity field? Should it not be 1/s (or 1/d) ?

Response: Corrected as suggested. (See Figure 13 in the revision).

**Baffin Bay sea ice inflow and outflow:1978/1979-2016/2017**

Haibo Bi1,2,3, Zehua Zhang1,2,3, Yunhe Wang1,2,4, Xiuli Xu1,2,3, Yu Liang1,2,4, Jue Huang5, Yilin Liu5, Min Fu,6Haijun Huang1,2,3,4

1Key laboratory of Marine Geology and Environment, Institute of Oceanology, Chinese Academy of Sciences, Qingdao, China
2Laboratory for Marine Geology, Qingdao National Laboratory for Marine Science and Technology, Qingdao, China
3Center for Ocean Mega-Science, Chinese Academy of Sciences, Qingdao, China

4University of Chinese Academy of Sciences, Beijing, China

5Shandong University of Science and Technology, Qingdao, China

[revised manuscript text omitted]
 final-derived Envisat estimates are smoothed-degraded to a 25-km grid, and then spatially registered to consistent with the NSIDC datagrids. NSIDC sea ice motion and SAR ice motion vectors in terms of drift speed ( $U_{NSIDC}$ ,  $U_{SAR}$ , in unit of km/d) and angle (or direction) are obtained at a given location and compared. The quality of the NSIDC is examined by the following scale: sea ice speed bias ( $U_{NSIDC}$ - $U_{SAR}$ ) (Figure 3a) and angular

20 (directional) difference of the sea ice drift vector (Figure 3b). Furthermore, IABP buoy measurements of daily mean sea ice motion from January 1979 to December 1994 (http://iabp.apl.washington.edu/) are used to assess the consistency of NSIDC-based sea ice area flux between the 1978-1987 and the later periods. There is at least 10 buoys in operation during 1979-1994 period in the Arctic Ocean. Overall comparisons in Figure 2 suggest that no significant difference with respect to the  $U_{NSIDC}$ - $U_{SAR}$  fields at different speed ranges is found between the two periods: 1979-1987 (Figure 2a) and 1988-1994

|------------------------|-------|-----|----|

25 (Figure 2b).